# Optimized Feature Generation for Tabular Data via LLMs with Decision Tree Reasoning

**Jaehyun Nam**[*1], **Kyuyoung Kim**[*1], **Seunghyuk Oh**[1]
**Jihoon Tack**[1], **Jaehyung Kim**[2], **Jinwoo Shin**[1]
[1]KAIST, [2]Yonsei University
{jaehyun.nam,kykim,jinwoos}@kaist.ac.kr

## Abstract

In tabular prediction tasks, tree-based models combined with automated feature engineering methods often outperform deep learning approaches that rely on learned representations. While these feature engineering techniques are effective, they typically depend on a pre-defined search space and primarily use validation scores for feature selection, thereby missing valuable insights from previous experiments. To address these limitations, we propose a novel tabular learning framework that utilizes large language models (LLMs), termed *Optimizing Column feature generator with decision Tree reasoning (OCTree)*. Our key idea is to leverage the reasoning capabilities of LLMs to identify effective feature generation rules without manually specifying the search space and provide language-based reasoning information highlighting past experiments as feedback for iterative rule improvements. We use decision trees to convey this reasoning information, as they can be easily represented in natural language, effectively providing knowledge from prior experiments (*i.e.*, the impact of the generated features on performance) to the LLMs. Our empirical results demonstrate that OCTree consistently enhances the performance of various prediction models across diverse benchmarks, outperforming competing automated feature engineering methods. Code is available at https://github.com/jaehyun513/OCTree.

## 1 Introduction

Learning useful representations from raw data is key to the success of deep learning algorithms, and their effectiveness has been demonstrated across multiple domains, *e.g.*, vision [1, 2, 3, 4] and language [5, 6]. However, in the tabular domain, deep learning approaches are often perceived as less effective [7, 8, 9, 10]. For instance, tree-based approaches utilizing raw column features of tabular data [11, 12] often outperform deep learning models in tabular prediction tasks such as classification and regression [13, 14, 15]. As a result, practitioners commonly resort to using tree-based methods coupled with manual feature engineering, such as computing the product of two column features [16].

Generating suitable column features, even with domain knowledge, can be challenging and costly. For instance, manual validation to identify useful features is infeasible due to the exponentially many possible combinations to explore [17]. To address this issue, existing feature engineering methods [17, 18, 19] use additional filtering schemes [20, 21, 22] to evaluate and select useful features automatically. While these approaches reduce manual effort and improve feature quality, they still present several challenges. First, practitioners often rely on manually defined search spaces to generate candidate features due to the inherent ambiguity of what constitutes informative features [17, 23]. However, this still requires substantial computation for validating candidate features,

---

[*]Equal contribution

38th Conference on Neural Information Processing Systems (NeurIPS 2024).

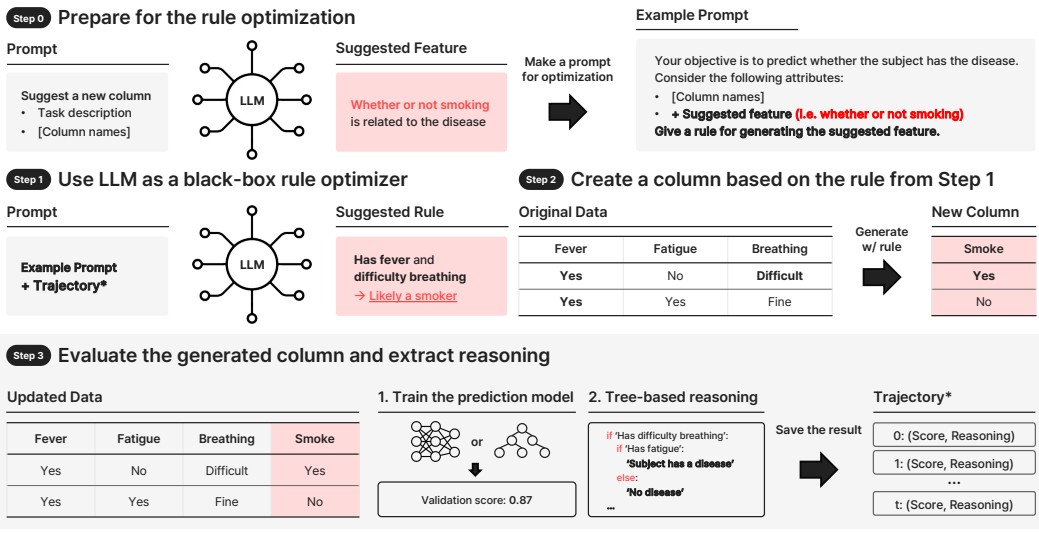

Figure 1: **Overview of OCTree.** (Step 0) Prompt the LLM to propose a name for the new column. (Step 1) Generate a rule by prompting the LLM with feedback on previously generated rules and relevant information for reasoning about the data. (Step 2) Generate a new column feature based on the proposed rule. (Step 3) Train a prediction model on the new data and compute the validation score and tree-based reasoning, provided as feedback for iterative improvements. (Step 4) Repeat steps 1-3 a fixed number of times and select the rule with the best validation score.

particularly as the number of features and the complexity of the search space grow. Furthermore, they neglect more effective experimental designs, relying solely on validation scores to select good features, despite the value of past experiment data for improving selection.

Motivated by this, we propose to approach this problem from a novel perspective: optimization to discover effective generation rules, leveraging the language understanding and reasoning capabilities of large language models (LLMs). Recent research has demonstrated that LLMs can optimize various non-differentiable problems using prompts that describe the optimization task in natural language [24, 25, 26]. This suggests the potential for LLMs to automatically generate and iteratively refine feature generators without the need for manually specifying the rule space. For example, the reasoning capabilities of LLMs allow incorporating feedback on their previous outputs into the process for iterative refinement. Moreover, linguistic contexts, such as column names (*e.g.*, 'Gender' and 'Age') and categorical values (*e.g.*, 'Female' and 'Male'), could be naturally integrated into the optimization [27, 28, 29, 30], which is difficult, if not impossible, with conventional methods.

**Contributions.** In this work, we leverage LLMs to generate novel column features for tabular prediction tasks, proposing *Optimizing Column feature generator with decision Tree reasoning (OCTree)*, a generic framework for automated feature generation using LLMs. Figure 1 illustrates an overview of our framework. Our approach begins by prompting an LLM to propose a name for a novel column feature based on the task description, such as 'Trading Volume' for stock price prediction. This initial suggestion guides the LLM in exploring and refining the corresponding feature values. Then, we further leverage the reasoning capability of LLMs to produce a *good* rule that generates values for the newly introduced column feature based on the existing ones. Specifically, starting from an initial rule $r_0$, we let the LLM iteratively improve the current rule $r_t$ by using extracted reasonings $d_0, d_1, \ldots, d_t$ and validation scores $s_0, s_1, \ldots, s_t$ attained by the prediction model as feedback. Here, $d_t$ denotes the language description of a decision tree fitted to the entire dataset, including the new feature generated by $r_t$. Specifically, we propose using the *decision tree reasoning* to provide the LLM with effective knowledge from the past experiments, *i.e.*, the prediction model trained with the generated features, providing learned knowledge about the entire dataset. This procedure is iterated for a fixed number of times, after which we select the rule with the highest validation score.

We assess the effectiveness of OCTree on a wide range of real-world datasets (*e.g.*, stock price and patient mortality prediction) from various sources, including recent Kaggle competitions. Our

experimental results demonstrate that OCTree consistently enhances the performance of various prediction models, including gradient-boosted decision trees [11] and deep neural networks [31, 32], for both classification and regression tasks. We also assess OCTree on datasets where language descriptions are unavailable, *i.e.*, all feature values and column names are anonymized during preprocessing. Even on these datasets, OCTree reduces relative prediction errors by an average of 5.0% compared to the best baseline model, *i.e.*, XGBoost on the 19 classification tasks benchmarked by Grinsztajn et al. [13]. Here, we use the Llama 2 7B model fine-tuned on high-quality dialogue data to enhance its ability to understand and generate contextually relevant, coherent rules. We also show that OCTree outperforms recent automatic feature engineering methods, including CAAFE [19] and OpenFE [17], often using our 7B LLM, even when these methods are combined with significantly more advanced models like GPT-4. Lastly, we demonstrate that features generated for one type of model (*e.g.*, a simpler model like XGBoost) can enhance the performance of other model types (*e.g.*, more complex models like neural networks). This illustrates a potential approach to scaling the method for larger, more complex models.

## 2 Related work

**Tabular learning with LLMs.** Recent developments in LLMs have encouraged investigation into their applications to tabular prediction tasks. Dinh et al. [27] and Hegselmann et al. [28] fine-tune GPT-3 [33] and T0 [34], respectively, by serializing tabular data into natural language. Nam et al. [35] utilizes unlabeled data expressed in natural language for few-shot semi-supervised tabular classification tasks via prompting LLMs. More recently, Yan et al. [36] introduced a tabular-specific tokenization method to pre-train a single language model capable of performing well across multiple tabular datasets. Instead of using LLMs as prediction models, we explore whether they can effectively generate useful column features for tabular prediction tasks. Specifically, we propose enhancing various prediction models by using LLMs as optimizers to generate novel column features.

**LLMs as optimizers.** Various prompting techniques have demonstrated the use of LLMs for solving optimization problems. This is achieved by describing optimization problems in natural language and instructing LLMs to iteratively generate new solutions based on previously found solutions and their evaluation. In particular, Yang et al. [25] uses LLMs to optimize linear regression, the traveling salesman problem, and prompt optimization (*i.e.*, refining instructions to improve LLM outputs). Building on these insights, we leverage LLMs to optimize feature generators for tabular prediction tasks. Unlike prior work, we incorporate decision tree reasoning as feedback, providing the model with learned knowledge about the dataset in natural language for more effective optimization.

**Automated feature engineering.** Automated feature engineering involves generating features from raw data without human effort to improve the performance on prediction tasks [19]. Various methods have been developed for this purpose [37, 38, 39], including iterative feature subsampling with beam search to select informative features [23] and feature boosting and pruning algorithms for efficient and accurate filtering [17]. More recently, Hollmann et al. [19] introduced a context-aware feature engineering approach that leverages LLMs to generate semantically meaningful features based on the description of a given task. Unlike previous approaches, our method leverages the optimization and reasoning capabilities of LLMs to discover effective feature generation rules without the need for manually defining a search space. Furthermore, while methods such as CAAFE [19] rely on language-based context, our approach is applicable to both context-aware and context-agnostic settings, enabling it to handle a broader range of prediction tasks.

## 3 Optimizing feature generator with decision tree reasoning

In this section, we introduce a framework for automated tabular feature engineering by leveraging the language understanding and reasoning capabilities of LLMs. In a nutshell, our approach utilizes LLMs as optimizers to propose and refine the rules for generating column features. Specifically, we iteratively improve the rules by guiding the LLMs using (1) the validation performance of previously proposed rules and (2) decision tree-based reasoning derived from the training data as inputs, enabling more effective optimization. We begin by framing the feature engineering problem as an optimization of the rules for generating column features (Section 3.1) and then introduce the core framework, termed *Optimizing Column feature generator with decision Tree reasoning* (OCTree), designed to solve this optimization task (Section 3.2).

**Problem setup.** Formally, the goal of tabular prediction tasks is to train a prediction model $f : \mathcal{X} \to \mathcal{Y}$, where $\mathcal{X}$ is the input space and $\mathbf{x} \in \mathcal{X}$ is an $M$-dimensional column feature with corresponding column names $\mathcal{C} = \{c_1, \ldots, c_M\}$. For example, '$x_1$: Female' and '$x_2$: 36' are values for the columns '$c_1$: Gender' and '$c_2$: Age', respectively. In classification tasks, $\mathbf{y} \in \mathcal{Y} = \{0,1\}^K$ represents the label space with $K$ classes, while in regression tasks, $\mathbf{y} \in \mathcal{Y} \subset \mathbb{R}$. We denote by $c_{\texttt{target}}$ the name of the column corresponding to the label $\mathbf{y}$.

## 3.1 Tabular feature generation as rule generator optimization

We frame tabular feature engineering as the optimization of feature-generating rules, where the rules define a mapping from the original set of features to a new feature. Our objective is to generate a one-dimensional column feature, $\mathcal{X}'$, by optimizing the rule $r : \mathcal{X} \to \mathcal{X}'$. Specifically, we aim to create a novel column feature that improves the performance of the prediction model when trained with the new feature, $f : \mathcal{X} \oplus \mathcal{X}' \to \mathcal{Y}$. Our optimization problem can be formalized as follows:

$$\min_r \mathcal{L}_{f^*}(\mathcal{D}_{\texttt{val}} \oplus r) \quad \text{subject to} \quad f^* = \arg\min_f \mathcal{L}_f(\mathcal{D}_{\texttt{train}} \oplus r), \tag{1}$$

where $g$ is the rule generator (*i.e.*, LLM $\mathcal{M}$ in our case), $r := g(\mathcal{D}_{\texttt{train}})$ is the rule generated based on the training dataset $\mathcal{D}_{\texttt{train}}$, and $\mathcal{D} \oplus r := \{\mathbf{x}_i \oplus r(\mathbf{x}_i), \mathbf{y}_i\}_{i=1}^N$ denotes the dataset augmented with the new column feature generated. $\mathcal{L}_f$ is the objective function for the given prediction task, such as mean absolute error for regression tasks, evaluated using the model $f$. In summary, we optimize the rule $r$ to achieve the best validation score measured on $\mathcal{D}_{\texttt{val}} \oplus r$ with the model $f^*$ trained to minimize the loss on $\mathcal{D}_{\texttt{train}} \oplus r$.

However, such bi-level optimization is often non-differentiable or computationally demanding, as it involves computing gradients through the optimization of $f^*$. Moreover, the rule itself may involve non-differentiable operations, such as logical conjunctions between categorical features (*e.g.*, 'Is a Smoker = Has Fever $\wedge$ Has Difficulty Breathing' as in Figure 1). One approach to addressing the issue is to use black-box optimization methods, such as evolutionary strategies [40] or reinforcement learning [41]. However, these methods also have limitations, including the need for a manually defined search space, which is complex, as well as the potentially suboptimal use of valuable feedback from previously proposed solutions that could enhance the optimization process.

To address this issue, we propose leveraging LLMs as optimizers for the tabular feature engineering problem. Our approach involves iteratively proposing and refining rules by prompting LLMs with a trajectory of feedback, which includes the history of previously proposed rules, the validation scores, and the associated reasoning information. Optimization using LLMs [25] has proven to be an effective tool, particularly for black-box optimization problems, which we show can also be effective in tabular feature engineering. Furthermore, LLMs can leverage the semantics of column and feature values for better optimization and provide the flexibility to operate without a pre-defined search space.

## 3.2 Generating column features with OCTree

We now present the core algorithm. First, we prompt the LLM to propose a name of a new column, such as 'Smoking Status' in Figure 1, and the corresponding rule based on the task description. We then compute the validation score of the prediction model and extract decision tree reasoning from the training dataset, initializing the optimization trajectory. This trajectory provides feedback to the LLM, with the *decision tree reasoning* component, which effectively conveys the knowledge of past experiments and captures the quality of previously suggested features. As the optimization process continues, the trajectory is updated, enabling the LLM to refine and enhance the rule iteratively.

**Column name generation.** OCTree begins by generating a name of a new column feature, $c_{\texttt{new}}$, through the LLM $\mathcal{M}$. This is done by prompting $\mathcal{M}$ with the prompt $p_{\texttt{col}}$ (see Appendix A.1), which asks for a new column name that would be useful for predicting the target: $c_{\texttt{new}} = \mathcal{M}(p_{\texttt{col}}(\mathcal{C}, c_{\texttt{target}}))$. Leveraging its language understanding capabilities, the LLM is able to generate semantically coherent and relevant column names. For instance, it might suggest using trading volume as a new feature for predicting stock prices.

**Initialize optimization and extract decision tree reasoning.** OCTree then generates an initial rule $r_0$ for deriving a new column feature from the original set of columns $\mathcal{C}$. This is done by prompting the LLM $\mathcal{M}$ with the prompt $p_{\texttt{init}}$ (see Appendix A.2) to propose a rule for predicting $c_{\texttt{new}}$ with

$c_1, \ldots, c_M$: $r_0 = \mathcal{M}(p_{\mathtt{init}}(\mathcal{C}, c_{\mathtt{new}}))$. The score $s_0$ for the initial rule is then evaluated with the prediction model $f^*$: $s_0 = \mathcal{L}_{f^*}(\mathcal{D}_{\mathtt{val}} \oplus r_0)$. Additionally, decision tree reasoning $d_0$ is extracted using CART [42] fitted to the training dataset:

$$d_0 = \mathtt{CART}(\mathcal{D}_{\mathtt{train}} \oplus r_0).$$

CART is a binary decision tree that recursively splits the data based on criteria such as Gini impurity to predict the outcome. We employ CART for two main reasons: (i) tree-based models, often ensembles of simple decision trees like CART, outperform deep learning on many tabular prediction tasks, and (ii) CART is easily interpretable and can be expressed in natural language. For example, as illustrated in Figure 1, CART can be expressed using a simple if-else syntax. Intuitively, the decision tree reasoning extracted by CART provides valuable insights learned from the entire training dataset. It explicitly highlights the columns that are considered more significant (as nodes in the tree) and the corresponding values (as thresholds of the nodes) used for prediction.

**Optimization with decision tree reasoning.** To optimize the rule, we describe the task in natural language and provide the trajectory $\mathcal{T}_t = \{(s_i, d_i, r_i)\}_{i=0}^t$, which includes the history of previously proposed rules, the corresponding scores, and associated reasoning information. We then generate a new rule $r_{t+1}$ using the LLM $\mathcal{M}$ with the prompt $p_{\mathtt{gen}}$ (see Appendix A.3), which asks $\mathcal{M}$ to propose a new rule that is not present in $\mathcal{T}_t$ and that would improve on the scores from the previous iterations: $r_{t+1} = \mathcal{M}(p_{\mathtt{gen}}(\mathcal{T}_t, \mathcal{C}, c_{\mathtt{target}}))$. The elements in the optimization trajectory are ordered by the scores, as LLMs tend to generate suggestions that appear later in the list [25, 43]. Afterward, we append the new score $s_{t+1} = \mathcal{L}_{f^*}(\mathcal{D}_{\mathtt{val}} \oplus r_{t+1})$, the decision tree reasoning $d_{t+1} = \mathtt{CART}(\mathcal{D}_{\mathtt{train}} \oplus r_{t+1})$, and the rule $r_{t+1}$ to the trajectory: $\mathcal{T}_{t+1} = \mathcal{T}_t \cup \{(s_{t+1}, d_{t+1}, r_{t+1})\}$. The optimization proceeds for a fixed number of iterations, with the best rule selected based on the highest validation score.

**Generating multiple features.** The optimization process can be repeated to generate multiple useful features. For example, after generating the column 'Smoking Status', an additional column 'Physical Activity Level' can be generated based on the original features and the newly created 'Smoking Status'. Formally, we first generate a new column $\mathcal{X}' = r_{\mathtt{opt}}(\mathcal{X})$, where $r_{\mathtt{opt}}$ is the optimized rule for generating the new feature $\mathcal{X}'$. This results in an augmented input space, $\mathcal{X}^{\mathtt{new}} = \mathcal{X} \oplus \mathcal{X}'$. Using the updated

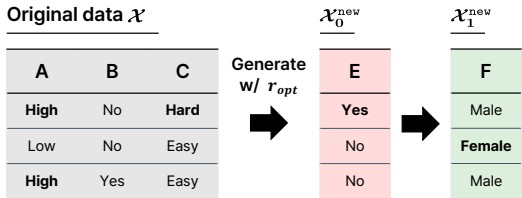

Figure 2: **Generation of multiple features.** The optimization process is repeated to generate multiple column features in sequence.

dataset $\mathcal{D}^{\mathtt{new}} \subseteq \mathcal{X}^{\mathtt{new}} \times \mathcal{Y}$, OCTree iteratively generates additional column features. This process continues, introducing new features in sequence until the validation score no longer improves.

# 4 Experiments

In this section, we evaluate the effectiveness of OCTree across a range of tabular classification and regression tasks using diverse datasets. Our findings demonstrate that OCTree consistently improves the performance of various types of prediction models (Section 4.1 and Section 4.2). Furthermore, ablation studies confirm the value of the proposed decision tree reasoning in the optimization and demonstrate that features generated using one type of prediction model can effectively transfer to others, suggesting an approach to scaling the framework to more complex models (Section 4.3).

**Datasets.** First, we select real-world datasets with language descriptions of the column features from diverse sources, including the Disease, Academic, Enefit, and Tesla Stock datasets, recently released on Kaggle, and the Clinical Trial dataset from the US National Library of Medicine. These prediction tasks are highly practical and relevant to domains such as healthcare (*e.g.*, diagnostics), academia (*e.g.*, student dropout prediction), and finance (*e.g.*, stock price forecasting). In addition, to evaluate OCTree on datasets without language descriptions of the columns, we include the 19 classification datasets benchmarked by Grinsztajn et al. [13]. When selecting datasets, it is important to consider the heterogeneous nature of tabular data [44] and ensure coverage of both categorical and numerical features, as well as both classification and regression tasks. We conduct experiments on this diverse selection of datasets to evaluate OCTree's general applicability across various types of tabular data. Further details on the datasets are provided in Appendix B.

Table 1: **Performance improvement by OCTree on datasets with language descriptions.** We report test error rates (%) for three classification tasks (*) and mean absolute error ($\times 10^{-3}$) for two regression tasks ($\dagger$). The lowest errors are highlighted in **bold**. Values in parentheses indicate the relative error rate reduction from the baseline. We report the mean error and standard deviation across three random splits, except for two regression tasks (time series tabular data), which are split by time index. N/A indicates that the method is not applicable, as HyperFast is a classification model.

| Method | LLM | Tesla$^\dagger$ | Enefit$^\dagger$ | Disease* | Clinical* | Academic* |
|--------|-----|------|--------|---------|----------|----------|
| | | | | *XGBoost* [11] | | |
| Baseline | - | 6.61 | 8.00 | $28.09_{\pm7.9}$ | $46.27_{\pm5.0}$ | $14.15_{\pm0.6}$ |
| **OCTree** | Llama 2 | 5.56 (15.9%) | 8.00 (0.0%) | $26.19_{\pm7.2}$ (6.8%) | $45.07_{\pm4.1}$ (2.6%) | $14.11_{\pm0.5}$ (0.3%) |
| **OCTree** | GPT-4o | **5.48 (17.1%)** | **7.82 (2.3%)** | $\mathbf{25.72_{\pm6.6}}$ **(8.4%)** | $\mathbf{43.75_{\pm4.4}}$ **(5.4%)** | $\mathbf{13.74_{\pm0.1}}$ **(2.9%)** |
| | | | | *MLP* [31] | | |
| Baseline | - | 7.41 | 33.53 | $38.10_{\pm3.6}$ | $41.77_{\pm1.7}$ | $14.41_{\pm0.8}$ |
| **OCTree** | Llama 2 | 5.23 (29.4%) | 29.99 (10.6%) | $32.86_{\pm5.7}$ (13.7%) | $39.80_{\pm2.3}$ (4.7%) | $14.26_{\pm0.7}$ (1.0%) |
| **OCTree** | GPT-4o | **5.01 (32.4%)** | **21.68 (35.3%)** | $\mathbf{30.95_{\pm5.8}}$ **(18.8%)** | $\mathbf{39.25_{\pm0.5}}$ **(6.0%)** | $\mathbf{14.22_{\pm0.5}}$ **(1.3%)** |
| | | | | *HyperFast* [32] | | |
| Baseline | - | N/A | N/A | $28.57_{\pm10.0}$ | $43.64_{\pm1.1}$ | $14.67_{\pm0.7}$ |
| **OCTree** | Llama 2 | N/A | N/A | $28.10_{\pm9.2}$ (1.6%) | $\mathbf{41.45_{\pm1.7}}$ **(5.0%)** | $14.49_{\pm0.5}$ (1.2%) |
| **OCTree** | GPT-4o | N/A | N/A | $\mathbf{27.14_{\pm3.8}}$ **(5.0%)** | $42.00_{\pm1.5}$ (3.8%) | $\mathbf{14.49_{\pm0.5}}$ **(1.2%)** |

**Baselines.** To validate our method, we evaluate it across three types of prediction models. We first consider XGBoost [11], a highly competitive tree-based model known for its effectiveness in the tabular domain. Second, we apply our method to multilayer perceptron (MLP; Gorishniy et al. [31]), the base architecture for deep learning models. Lastly, we show that OCTree improves the performance of HyperFast [32], a recently introduced model designed for fast classification of tabular data. Implementation details, including hyperparameter search space, are provided in Appendix C.

**Common setup.** For all datasets, 60% of the data is used for training, 20% for validation, and 20% for testing. Following Gorishniy et al. [31], we use learned embeddings for categorical features when training MLPs, while HyperFast handles categorical features automatically. For all experiments, we use CART with a maximum depth of 4 to extract decision tree reasoning, provided to the rule generating LLM in the prompt. Unless noted otherwise, we use the Llama 2 model [45] at the 7B scale, fine-tuned on UltraChat [46], a dialogue dataset that has been used to develop strong chat models such as UltraLM [46]. Our findings indicate that open models, even at moderate scales, can be highly effective, particularly when equipped with strong chat capabilities. Further comparisons across different types of LLMs are provided in Section 4.2.

## 4.1 Main results: Context-aware feature engineering

**Datasets with language descriptions.** We first experiment with datasets where language descriptions of the columns and categorical feature values are available. In these cases, the LLM generates a logical rule in *natural language*, which is then converted into Python code for use in our experiments. For this task, we utilize both GPT-4o and our Llama 2 model, as the Llama 2 model can be limited by its relatively short context length on some datasets. See Appendix A.4 for the prompt used.

As shown in Table 1, OCTree consistently enhances the performance of various baseline models. For instance, when generating the column 'Trading Volume' for the Tesla Stock dataset using Llama 2 for XGBoost, the relative error is reduced by 15.9%. OCTree is compatible with arbitrary LLMs, allowing more advanced models to enhance the quality of generated features further. Specifically, with GPT-4o, one of the latest LLMs from OpenAI, our method achieves a relative error reduction of 17.1% on the same dataset for XGBoost. We provide results on additional datasets in Table 13.

**Comparison with CAAFE.** CAAFE [19] also introduces a feature engineering approach that utilizes LLMs to construct features based on the linguistic context. However, it is important to note that CAAFE requires explicit language descriptions of the features, which limits its applicability when such information is unavailable. For example, feature names and values are often obfuscated for confidentiality, a common practice in the financial and medical domains. In contrast, our method can be effectively applied to datasets without linguistic descriptions, as demonstrated in Table 3.

Table 2: **Applicability and comparison of automated feature engineering methods.** We report the mean error (%) and standard deviation across the six datasets with language descriptions used in Tables 1 and 13. The lowest error is highlighted in **bold**. Values in parentheses indicate the relative error rate reduction from the baseline model (*i.e.*, XGBoost [11]), while N/I indicates no gain.

| Method | Applicability | | Comparison | |
| | w/o descriptions | w/ descriptions | LLM | Avg. Err. (%) |
| --- | --- | --- | --- | --- |
| Baseline | - | - | - | $25.87_{\pm 2.2}$ |
| AutoFeat [23] | ✓ | ✗ | - | $25.76_{\pm 2.1}$ (0.4%) |
| OpenFE [17] | ✓ | ✗ | - | $26.44_{\pm 1.7}$ ( N/I ) |
| CAAFE [19] | ✗ | ✓ | GPT-4o | $25.43_{\pm 2.2}$ (1.7%) |
| **OCTree (Ours)** | ✓ | ✓ | Llama 2 | $25.12_{\pm 1.9}$ (2.9%) |
| | | | GPT-4o | $\mathbf{24.53}_{\pm 1.9}$ **(5.2%)** |

Thus, to compare CAAFE with OCTree, we evaluate both methods on datasets with contextual information, particularly all of the six classification datasets used in Tables 1 and 13. The results in Table 2 show that our method significantly outperforms CAAFE with GPT-4o, even when using our custom Llama 2 model fine-tuned on open dialogue data. Also, conventional feature engineering methods, such as OpenFE [17], often struggle to generate meaningful features, particularly due to the difficulty of applying arithmetic operations to categorical features. A key distinction between CAAFE and OCTree is that our approach generates more semantically meaningful column names, which serve as a basis for creating high-quality features. Leveraging the LLM's reasoning and in-context learning capabilities, we guide the model in effectively navigating the feature space to generate coherent, relevant rules, using a history of feedback on candidate features and decision tree reasoning to enhance its understanding of the data. In contrast, CAAFE primarily relies on language understanding to suggest simple combinations of existing feature. Moreover, CAAFE tends to adopt a greedy approach, evaluating the validation score for a candidate feature only once and discarding it if no improvement is observed. We provide further results with case studies in Appendix E.

## 4.2 Main result: Context-agnostic feature engineering

**Datasets without language descriptions.** In practice, datasets do not always include clear language descriptions of the prediction task. For example, feature names and values in financial datasets are often obfuscated with arbitrary symbols to protect confidentiality [47]. OCTree adapts easily to such datasets without language descriptions and can generate features using various arithmetic rules. As shown in our ablations in Section 4.3, this is due to the more effective use of LLMs' optimization capabilities with decision tree reasoning that enhances the model's understanding of the data.

To evaluate datasets without language descriptions, we apply ordinal encoding to categorical features and normalize all features using a min-max scaler, transforming the original numeric values. We also use non-descriptive column names, such as $\mathcal{C} = \{\text{'x1'}, \text{'x2'}, \ldots, \text{'x5'}\}$ for a dataset with $M = 5$ columns. To initialize the feedback trajectory, we create an initial rule that is the product of the two columns with the highest importance weights computed using an XGBoost model, *e.g.*, $x_6 = x_1 \times x_5$. As shown in Table 3, our framework enhances baseline models even in the absence of language descriptions, achieving an average error reduction of approximately 5.0% for both the XGBoost classifier and MLP. For HyperFast, OCTree also improves the test error on 16 out of 19 datasets.

**Analysis of experiments with various open LLMs.** To evaluate how our method performs with various types of LLMs, we assess the performance of several open LLMs as rule generators. As shown in Table 4, while all models yield improvements over the baseline, we find our own model (*i.e.*, Llama 2 fine-tuned on UltraChat) to be particularly effective. This shows that our framework can be effectively implemented even with open models at a moderate scale, especially those with sufficiently strong chat capabilities. We suspect that this improvement stems from the enhanced ability of these models to understand and generate contextually relevant and

Table 4: **OCTree with Llama 2 variants.** We report the average test error rates (%) and standard deviations across three random seeds on the 19 datasets without language descriptions.

| Method | LLM | Size | Avg. Err. |
| --- | --- | --- | --- |
| XGBoost | - | - | $16.53_{\pm 0.1}$ |
| **OCTree** | Llama 2 Chat | 7B | $16.32_{\pm 0.1}$ |
| **OCTree** | Code Llama | 7B | $15.83_{\pm 0.2}$ |
| **OCTree** | **Ours** | 7B | $\mathbf{15.71}_{\pm 0.4}$ |

Table 3: **Performance improvement by OCTree on datasets without language descriptions.** We report test error rates (%) on the 19 classification tasks from Grinsztajn et al. [13]. The lowest error is in **bold**. Values in parentheses indicate the relative error rate reduction from the baseline, while N/I indicates no gain. We report the mean error and standard deviation across the three random splits.

| | XGBoost | | MLP | | HyperFast | |
|---|---|---|---|---|---|---|
| Dataset | Baseline | OCTree (Ours) | Baseline | OCTree (Ours) | Baseline | OCTree (Ours) |
| electricity | $8.32_{\pm0.0}$ | **$6.65_{\pm0.1}$ (20.1%)** | $15.64_{\pm0.3}$ | **$14.82_{\pm0.4}$ ( 5.2%)** | $15.25_{\pm0.5}$ | **$14.70_{\pm0.5}$ ( 3.6%)** |
| rl | $23.61_{\pm0.8}$ | **$19.32_{\pm0.4}$ (18.2%)** | $32.03_{\pm4.2}$ | **$28.30_{\pm1.7}$ (11.6%)** | $33.77_{\pm1.3}$ | **$33.50_{\pm1.2}$ ( 0.8%)** |
| compass | $22.91_{\pm0.5}$ | **$18.89_{\pm0.4}$ (17.6%)** | $27.41_{\pm1.0}$ | **$26.78_{\pm0.1}$ ( 2.3%)** | $25.74_{\pm0.6}$ | **$24.91_{\pm1.1}$ ( 3.2%)** |
| covertype | $9.10_{\pm0.2}$ | **$7.96_{\pm0.0}$ (12.5%)** | $8.73_{\pm0.4}$ | **$8.25_{\pm0.3}$ ( 5.5%)** | $9.86_{\pm1.6}$ | **$9.21_{\pm1.3}$ ( 6.6%)** |
| phoneme | $10.89_{\pm0.5}$ | **$10.15_{\pm0.7}$ ( 6.8%)** | $12.06_{\pm0.8}$ | **$10.98_{\pm0.6}$ ( 9.8%)** | **$10.55_{\pm0.7}$** | $10.57_{\pm0.9}$ ( N/I ) |
| kddCup09 | $19.86_{\pm1.1}$ | **$19.07_{\pm1.4}$ ( 4.0%)** | $24.30_{\pm0.3}$ | **$24.30_{\pm1.6}$ ( 0.0%)** | $25.75_{\pm0.7}$ | **$24.46_{\pm1.1}$ ( 5.0%)** |
| pol | $1.69_{\pm0.2}$ | **$1.62_{\pm0.2}$ ( 4.0%)** | $1.37_{\pm0.3}$ | **$1.27_{\pm0.3}$ ( 7.3%)** | $1.70_{\pm0.4}$ | **$1.55_{\pm0.2}$ ( 8.8%)** |
| Magic | $14.25_{\pm0.3}$ | **$13.75_{\pm0.4}$ ( 3.5%)** | $14.60_{\pm0.2}$ | **$14.50_{\pm0.0}$ ( 0.7%)** | $14.95_{\pm0.2}$ | **$14.34_{\pm0.5}$ ( 4.1%)** |
| california | $9.45_{\pm0.6}$ | **$9.13_{\pm1.0}$ ( 3.4%)** | $11.91_{\pm0.5}$ | **$11.37_{\pm0.1}$ ( 4.5%)** | $11.75_{\pm0.7}$ | **$11.02_{\pm0.6}$ ( 6.2%)** |
| house_16H | $11.66_{\pm0.5}$ | **$11.32_{\pm0.2}$ ( 3.0%)** | $13.07_{\pm0.2}$ | **$12.54_{\pm0.6}$ ( 4.1%)** | $12.77_{\pm0.3}$ | **$12.29_{\pm0.4}$ ( 3.8%)** |
| eye_movements | $35.06_{\pm0.7}$ | **$34.17_{\pm2.0}$ ( 2.6%)** | $40.03_{\pm1.2}$ | **$39.86_{\pm1.9}$ ( 0.4%)** | $41.33_{\pm1.5}$ | **$40.29_{\pm1.7}$ ( 2.5%)** |
| road-safety | $21.14_{\pm0.0}$ | **$20.65_{\pm0.1}$ ( 2.3%)** | $22.17_{\pm0.4}$ | **$21.87_{\pm0.1}$ ( 1.4%)** | $24.54_{\pm0.3}$ | **$24.07_{\pm0.4}$ ( 1.9%)** |
| kdd_ipums_la | $10.89_{\pm1.0}$ | **$10.69_{\pm1.0}$ ( 1.8%)** | $13.13_{\pm1.3}$ | **$11.72_{\pm1.5}$ (10.7%)** | $16.15_{\pm0.3}$ | **$13.55_{\pm1.4}$ (16.1%)** |
| MiniBooNE | $5.48_{\pm0.2}$ | **$5.42_{\pm0.1}$ ( 1.2%)** | $9.69_{\pm0.3}$ | **$7.35_{\pm0.2}$ (24.1%)** | $6.61_{\pm0.4}$ | **$6.54_{\pm0.2}$ ( 1.1%)** |
| credit | $22.02_{\pm0.3}$ | **$21.78_{\pm0.3}$ ( 1.1%)** | $24.43_{\pm0.6}$ | **$23.23_{\pm0.7}$ ( 4.9%)** | $25.06_{\pm1.1}$ | **$24.30_{\pm1.8}$ ( 3.0%)** |
| Higgs | $27.95_{\pm0.7}$ | **$27.91_{\pm0.2}$ ( 0.1%)** | $29.43_{\pm0.6}$ | **$28.80_{\pm0.2}$ ( 2.1%)** | $30.04_{\pm0.2}$ | **$29.73_{\pm0.5}$ ( 1.0%)** |
| jannis | **$20.61_{\pm0.1}$** | $20.64_{\pm0.1}$ ( N/I ) | **$22.28_{\pm0.1}$** | $22.51_{\pm0.1}$ ( N/I ) | $24.29_{\pm0.4}$ | **$23.65_{\pm0.3}$ ( 2.6%)** |
| wine | **$19.11_{\pm3.3}$** | $19.18_{\pm3.9}$ ( N/I ) | **$21.53_{\pm3.1}$** | $21.59_{\pm1.4}$ ( N/I ) | **$19.18_{\pm2.7}$** | $19.31_{\pm2.2}$ ( N/I ) |
| bank-marketing | **$20.09_{\pm0.3}$** | $20.31_{\pm0.6}$ ( N/I ) | $21.11_{\pm0.4}$ | **$21.09_{\pm0.4}$ ( 0.1%)** | $21.25_{\pm1.0}$ | $21.66_{\pm0.8}$ ( N/I ) |

Table 5: **Comparison with automatic feature engineering methods.** We report the mean error (%) and standard deviation across the 22 datasets used in Tables 1 and 3. The lowest error is highlighted in **bold**, and the second lowest is underlined. Values in parentheses indicate the relative error rate reduction from the baseline model. OCTree$^\dagger$ refers to our method integrated with other approaches.

| Prediction model | Baseline | AutoFeat | OpenFE | OCTree (Ours) | OCTree$^\dagger$ (Ours) |
|---|---|---|---|---|---|
| XGBoost | $18.30_{\pm0.3}$ | $18.24_{\pm0.3}$ (1.3%) | $17.79_{\pm0.2}$ (2.8%) | $17.45_{\pm0.5}$ (4.6%) | **$16.85_{\pm0.3}$ (7.9%)** |
| MLP | $20.88_{\pm0.1}$ | $20.60_{\pm0.5}$ (1.3%) | $20.12_{\pm0.5}$ (3.6%) | $19.91_{\pm0.4}$ (4.6%) | **$19.41_{\pm0.5}$ (7.0%)** |

coherent rules, resulting in better optimization outcomes. Specifically, as illustrated by the examples in Appendix F, our model navigates a broader space of features more effectively than the base Llama 2 chat model and demonstrates the ability to utilize built-in Python functions such as 'abs()'. Code Llama also demonstrates strong performance due to its training on code data that includes a variety of arithmetic rules, which is especially useful for generating rules for datasets lacking language descriptions. Notably, the model can leverage a range of NumPy operations (*e.g.*, 'np.sin'), allowing it to produce more mathematically complex features. This suggests that LLMs trained on high-quality code and dialogue data could serve as even more effective rule generators within our framework.

### 4.3 Ablations and analysis

**Integrating with other automated feature engineering methods.** OCTree is complementary to existing feature engineering methods, allowing for integration in several natural ways. One simple approach is to first apply OCTree to generate features, followed by the use of other methods to further refine and augment the feature set. In Table 5, we first compare the standalone performance of OCTree, demonstrating that it outperforms competing state-of-the-art automated feature engineering methods (*i.e.*, AutoFeat [23] and OpenFE [17]) for both XGBoost and MLP models. Moreover, combining OCTree with OpenFE (denoted as OCTree$^\dagger$ in Table 5) further boosts performance, achieving a 7.9% reduction in relative error for XGBoost.

**Ablation study on the proposed components.** As shown in Table 6, our framework consists of two essential components: (i) generating new column features (denoted as *Gen. Feat.* in Table 6), and (ii) providing explicit decision tree reasoning as feedback (denoted as *DT reasoning* in Table 6) during the optimization process. First, note that the rules are sufficiently well-optimized even without an explicit decision tree provided as feedback; the LLM improves performance based solely on score

Table 6: **Ablation study of the proposed decision tree reasoning.** We report the mean error (%) and standard deviation across three random splits on two datasets with language descriptions (*) and two datasets without language descriptions ($\dagger$). The lowest error is highlighted in **bold**. Values in parentheses indicate the relative error rate reduction from the baseline model.

| Gen. Feat. | DT Reasoning | Disease* | Clinical* | electricity$^\dagger$ | kddCup09$^\dagger$ |
|---|---|---|---|---|---|
| - | - | $28.09_{\pm7.9}$ | $46.27_{\pm5.0}$ | $8.32_{\pm0.0}$ | $19.86_{\pm1.1}$ |
| ✓ | ✗ | $27.62_{\pm8.4}$ (1.7%) | $45.61_{\pm4.1}$ (1.4%) | $6.89_{\pm0.6}$ (17.2%) | $19.47_{\pm1.6}$ (2.0%) |
| ✓ | ✓ | $\mathbf{26.19_{\pm7.2}}$ **(6.8%)** | $\mathbf{45.07_{\pm4.1}}$ **(2.6%)** | $\mathbf{6.65_{\pm0.1}}$ **(20.1%)** | $\mathbf{19.07_{\pm1.4}}$ **(4.0%)** |

Table 7: **Performance improvement through feature transfer.** We optimize the feature generation rule using XGBoost and transfer the generated features to improve MLP and HyperFast (OCTree$_\texttt{trans}$). We report the test error rates (%) and standard deviation across three random seeds for two datasets with language descriptions (*) and two datasets without ($\dagger$). The lowest error is in **bold**, with values in parentheses indicating the relative error rate reduction from the baseline model. N/I denotes cases where no improvement was observed.

| | MLP | | HyperFast | |
|---|---|---|---|---|
| Dataset | Baseline | OCTree$_\texttt{trans}$ **(Ours)** | Baseline | OCTree$_\texttt{trans}$ **(Ours)** |
| Disease* | $38.10_{\pm3.6}$ | $\mathbf{35.24_{\pm4.4}}$ **(7.5%)** | $28.57_{\pm10.0}$ | $\mathbf{27.62_{\pm5.8}}$ **(5.8%)** |
| Clinical* | $\mathbf{41.77_{\pm1.2}}$ | $42.32_{\pm2.3}$ ( N/I ) | $43.64_{\pm1.1}$ | $\mathbf{42.76_{\pm1.8}}$ **(2.0%)** |
| electricity$^\dagger$ | $15.64_{\pm0.3}$ | $\mathbf{15.03_{\pm0.3}}$ **(3.9%)** | $15.37_{\pm0.4}$ | $\mathbf{14.88_{\pm0.2}}$ **(3.2%)** |
| kddCup09$^\dagger$ | $24.30_{\pm0.3}$ | $\mathbf{23.47_{\pm0.5}}$ **(3.4%)** | $25.62_{\pm0.7}$ | $\mathbf{25.22_{\pm0.9}}$ **(1.6%)** |

feedback. However, providing the decision tree as feedback to the LLM can lead to even better performance. We believe that decision tree reasoning, which highlights important columns and their threshold values, enables the LLM to understand the data better, resulting in the generation of more contextually relevant and useful rules. Moreover, decision trees can be easily represented in natural language using if-else syntax, effectively conveying the information about the data to the LLM.

**Transferring generated rules to other prediction models.** While we optimize feature generation rules to improve the performance of a specific prediction model, the generated features can also be utilized in other models to achieve similar improvements. For example, it would be more efficient to generate features using XGBoost, which typically trains and evaluates faster than larger deep neural networks, and then apply these features to more complex models. To assess whether such feature transfer is feasible within our framework, we first optimize the column generation rules using XGBoost and then train MLP and HyperFast models with the generated features. As shown in Table 7, these features significantly improve the performance of both models, demonstrating the effectiveness of this approach, especially when only limited computational resources are available.

**Evaluating the validity of generated features.** The rule generator LLM is asked to recommend a new column feature that is not already present in the dataset. Here, we evaluate whether the LLM is capable of generating features that are, in fact, valid and relevant to the target task. We perform this analysis from two perspectives: (i) whether the LLM can identify the most relevant column feature when provided with multiple candidate columns, and (ii) whether using real-world data, when available, for the suggested column leads to improved performance of prediction models.

Our method is based on the assumption that sufficiently capable LLMs can understand the relationship between the target task and column features, enabling them to generate new, relevant features for the task. To evaluate this assumption, we first examine whether the LLMs can identify the features that are more relevant to the prediction task. For this experiment, we begin by removing two existing features from a dataset and then prompt the LLM to rank the two features according to their importance for the target task. Specifically, we remove 'Cholesterol Level' and 'Cough' from the original Disease dataset and then ask the LLM to identify the attribute more relevant to predicting whether a patient has a disease. Both the GPT-4o and Llama 2 models indicate that 'Cough' is more important than 'Cholesterol Level'. As shown in Table 8, XGBoost achieves a lower error rate when trained with the 'Cough' feature compared to when trained with 'Cholesterol Level', which is consistent with the LLMs' assessment that the former is more relevant to the task.

Table 8: **LLM identifies important features.** We report the mean error (%) and standard deviation across three random splits on the Disease dataset. Both GPT-4o and Llama 2 identify the cough feature as more important, consistent with the accuracy seen in XGBoost models trained with and without these features.

| Column feature | | Model |
|---|---|---|
| Cough | Cholesterol | XGBoost |
| ✗ | ✗ | $34.76_{\pm0.8}$ |
| ✗ | ✓ | $33.34_{\pm0.8}$ |
| ✓ | ✗ | $30.00_{\pm4.3}$ |
| ✓ | ✓ | $28.09_{\pm7.9}$ |

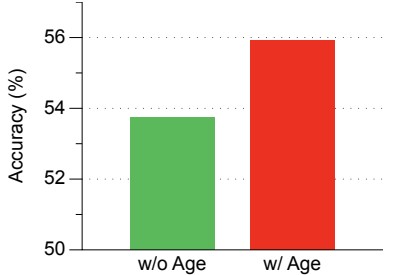

Figure 3: **Imputing features with real data, *i.e.*, *Age*.** We report the mean accuracy (%) across three random splits on the Clinical dataset using XGBoost.

Building on the LLM's ability to identify important features for the target task, we further assess whether the generated columns can be imputed with real-world data to enhance prediction. In this experiment, we use the Clinical Trial dataset, where the LLM introduced the column 'Age' when prompted to suggest a new column. We then incorporated real-world age data from the US National Library of Medicine to augment the original dataset. As shown in Figure 3, training on this augmented data results in a notable improvement, demonstrating that the LLM-generated columns align well with real-world data. In conclusion, we recommend that practitioners utilize OCTree to either (i) identify additional column features and collect the corresponding real-world data or (ii) optimize feature generation rules in the absence of real-world data.

**Analysis of the rule optimization process.** We analyze how the output rules evolve throughout the optimization rounds on the electricity dataset. Due to space constraints, we show the first and last five output rules in Appendix H. In the early stages of optimization, the LLM generates a diverse range of outputs, indicating an active exploration of potential rules. In contrast, during the later stages, the LLM focuses on refining the solution space around previously identified rules, making only minor adjustments. This again demonstrates that with appropriate guidance through the optimization process, sufficiently capable LLMs can serve as highly effective optimizers.

**Handling hallucinations.** While LLMs may occasionally suggest suboptimal or semantically incoherent rules, our method is designed to address these hallucinations. Specifically, we provide feedback on previously generated rules to guide the LLMs in iteratively improving the rule generation. This feedback loop helps the LLMs avoid hallucinations that might lead to low validation scores in subsequent iterations. Empirically, we find that these issues are more common in the early stages when the LLMs explore the rule space more broadly and in less capable models, *e.g.*, those without additional training on dialogue or code generation data.

## 5 Conclusion

In this paper, we propose OCTree, a generic framework that leverages the power of LLMs (*e.g.*, reasoning capability) for automatically generating column features for tabular prediction tasks. We evaluate the effectiveness of OCTree across various prediction tasks and demonstrate that our method consistently enhances the performance of diverse prediction models, often significantly more effectively than competing feature engineering methods. As future work, exploring feedback-based alignment methods, such as reinforcement learning from human feedback, to further enhance LLMs as rule generators would be an exciting direction to explore.

**Limitation.** One potential limitation of our work is that evaluating the generated features involves computing the validation scores of the prediction model, which can be time-consuming if the model requires extensive training. However, as demonstrated by the results in Table 7, this issue can be mitigated by first generating features with a simpler prediction model and then transferring those features to the target model, reducing the overall runtime and computational requirements.

## Acknowledgements and disclosure of funding

We would like to thank Sihyun Yu, Subin Kim, Changyeon Kim, Daewon Choi, Jinseong Park, and anonymous reviewers for their helpful feedback and discussions. This work was supported by Institute for Information & communications Technology Promotion (IITP) grant funded by the Korea government (MSIT) (No.RS-2019-II190075, Artificial Intelligence Graduate School Program (KAIST); No.RS-2022-II220184, 2022-0-00184, Development and Study of AI Technologies to Inexpensively Conform to Evolving Policy on Ethics), the NIPA (National IT Industry Promotion Agency), through the Ministry of Science and ICT (Hyperscale AI flagship project), and Hyundai Motor Group.

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

# Appendix: Optimized Feature Generation for Tabular Data via LLMs with Decision Tree Reasoning

## A  Prompt examples

### A.1  Generation of a new column

In Listing 1, we present the prompt $p_{col}$, which instructs the LLM to generate a new column name $c_{new}$. The prompt includes a detailed explanation of each feature column, specifying its type and values. For convenience, we restricted $c_{new}$ to be either binary or categorical.

```
f'''### Your task ###
Your objective is to predict {Objective}. You have access to the following
↪  attributes:
- {Column 1 Name}: (Numerical value of {Column 1 Min} ~ {Column 1 Max})
- {Column 2 Name}: (Boolean)
- {Column 3 Name}: (Categorical value of {Value 1}, {Value 2})
...

To enhance prediction performance, what additional attributes should be
↪  considered? These attributes should be either binary (e.g., 'Yes' or 'No') or
↪  categorical (e.g. 'high', 'low', or 'moderate'). Please propose a new
↪  attribute that is not listed above.

### Answer ###
'''
```

Listing 1: Prompt for the generation of a new column $c_{new}$.

### A.2  Rule initialization

In Listing 2, we present the prompt $p_{init}$, which instructs the LLM to create an initial rule for generating a new column $c_{new}$. The LLM generates a new column, $c_{new}$, by considering the existing column features in the dataset and the semantic meaning of the name of $c_{new}$.

```
f'''### Your task ###
You have access to the following attributes:
- {Column 1 Name}: (Numerical value of {Column 1 Min} ~ {Column 1 Max})
- {Column 2 Name}: (Boolean)
- {Column 3 Name}: (Categorical value of {Value 1}, {Value 2}, {Value 3})
...

### Question ###
Give a good rule to predict the '{New Column Name}' ({Output Category 1} or
↪  {Output Category 2}) with the attributes listed above.

### Answer ###
'''
```

Listing 2: Prompt for the rule initialization.

## A.3 Generation of a rule

In Listing 3, we present the prompt $p_{\text{gen}}$, which instructs the LLM to generate an improved rule compared to those previously generated. The prompt includes a list of (rule, tree-based reasoning, score) tuples. Since the score represents the accuracy of the XGBoost classifier, the trajectory is sorted in ascending order of score. For regression tasks using mean absolute error, the trajectory is sorted in descending order.

```
f'''I have some rules to predict {Objective} with attributes listed below.
- {Column 1 Name} (Numerical value of {Column 1 Min} ~ {Column 1 Max}): {Column 1
↪  Feature Description}
- {Column 2 Name} (Boolean): {Column 2 Feature Description}
- {Column 3 Name} (Categorical value of {Value 1}, {Value 2}, {Value 3}): {Column
↪  3 Feature Description}
...

We also have corresponding decision trees (CART) to predict {Objective} from the
↪  attributes listed above along with predicted {New Column Name}.
The rules are arranged in ascending order based on their scores evaluated with
↪  XGBoost classifier, where higher scores indicate better quality.

Rule to predict {Objective}:
{Rule 1}
Decision tree (CART):
{Decision Tree 1}
Score evaluated with XGBoost classifier:
{Score 1}

Rule to predict {Objective}:
{Rule 2}
Decision tree (CART):
{Decision Tree 2}
Score evaluated with XGBoost classifier:
{Score 2}

...

Give me a new rule to predict {Objective} that is different from the old ones (but
↪  should use the listed attributes above) and has a score as high as possible.

Improved rule:'''
```

Listing 3: Prompt for the rule generation.

## A.4 Translation of a rule into Python code

In Listing 4, we present the prompt $p_{\text{code}}$, which instructs the LLM to translate the rule into Python code. To minimize potential syntactic errors, we impose several restrictions. First, we explicitly define the variable types wherever possible. We also instruct the LLM to account for the feature value types before performing calculations, *e.g.*, to avoid the addition of categorical and numerical values.

```
f'''### Rule ###
{Rule to translate}

### Your Task ###
Change the rule into executable Python code. Consider the type of each feature.
Input data (Numpy array (not a Dict)): [{all_column_names}]
- {Column 1 Name}: (Numerical value of {Column 1 Min} ~ {Column 1 Max})
- {Column 2 Name}: (Boolean)
- {Column 3 Name}: (Categorical value of {Value 1}, {Value 2})
...

Output: {Output Category 1} or {Output Category 2}
Give instantly executable code without example usage. Consider the feature value
↪   types (avoid calculating categorical value and numerical value). Start with
↪   'def {Rule Function Name}(data):'

### Only python code ###
'''
```

Listing 4: Prompt for translating rule in natural language into Python code.

# B Dataset details

In this section, we provide further details on the datasets.

## B.1 Datasets with language descriptions

### 1. Disease[1]

A classification task to predict a patient's disease diagnosis based on the following attributes:

- **Binary (Yes or No)**: Fever, Fatigue
- **Numerical (Range)**: Age ($25 \sim 90$)
- **Categorical**: Gender (Female or Male), Blood Pressure (High, Normal, or Low), Cholesterol Level (High, Normal, or Low)

### 2. Clinical Trial[2]

A classification task to predict patient mortality in clinical trials based on the following attributes:

- **Binary (Yes or No) - Historical Disease**: Deep Vein Thrombosis, Pulmonary Embolism, Antiandrogen Therapy, Cardiac Failure, Respiratory Failure, Venous Insufficiency, Coronary Artery Disease, Myocardial Infarction, Hypertension, Peripheral Arterial Occlusive Disease
- **Binary (Yes or No) - Medication**: Dexamethasone, Ondansetron, Heparin, Fluorouracil, Ranitidine, Cisplatin, Metoclopramide, Carboplatin, Furosemide

### 3. Academic[3]

A classification task to predict whether a student would dropout based on the following attributes:

- **Numerical**: Marital Status, Daytime/Evening Attendance, Previous Qualification, Nationality, Father's Qualification, Father's Occupation, Displaced, Debtor, Tuition Fees up to Date, Gender, Scholarship Holder, Age at Enrollment, International, Curricular Units 1st Sem (Approved), Curricular Units 1st Sem (Grade), Curricular Units 2nd Sem (Approved), Curricular Units 2nd Sem (Grade).

### 4. Enefit[4]

A regression task to predict daily energy consumption based on the following attributes:

- **Numerical**: Prediction Unit Id, Day, Hour, Lowest Price Per MWh, Highest Price Per MWh, Installed Capacity, Euros Per MWh, Local Forecast Temperature, Local Forecast Dewpoint, Local Forecast Cloudcover Total, Local Forecast 10 Metre U Wind Component, Local Forecast 10 Metre V Wind Component, Local Forecast Direct Solar Radiation, Local Forecast Surface Solar Radiation Downwards, Local Forecast Total Precipitation.

### 5. Tesla Stock[5]

A regression task to predict the target day's highest stock price based on the following attributes:

- **Numerical**: Open Price of 2 Days Before, Highest Price of 2 Days Before, Lowest Price of 2 Days Before, Close Price of 2 Days Before, Open Price of 1 Day Before, Highest Price of 1 Day Before, Lowest Price of 1 Day Before, Close Price of 1 Day Before, Open Price of the Target Day, Time Index.

---

[1] https://www.kaggle.com/datasets/uom190346a/disease-symptoms-and-patient-profile-dataset
[2] https://data.projectdatasphere.org/projectdatasphere/html/content/119
[3] https://www.kaggle.com/datasets/missionjee/students-dropout-and-academic-success-dataset
[4] https://www.kaggle.com/competitions/predict-energy-behavior-of-prosumers
[5] https://www.kaggle.com/datasets/guillemservera/tsla-stock-data

## B.2 Datasets without language descriptions

For our main results on datasets without language descriptions (see Table 3), we evaluate the 19 classification datasets from the tabular benchmark proposed by Grinsztajn et al. [13]. Following the curation approach from Grinsztajn et al. [13]), we uniformly subsample to 50,000 instances for datasets exceeding this size. We provide brief dataset statistics below.

Table 9: **Dataset statistics.** 19 classification datasets benchmarked by Grinsztajn et al. [13].

| Dataset | OpenML ID | # Samples | # Features |
|---|---|---|---|
| rl | 44160 | 4970 | 12 |
| electricity | 44156 | 38474 | 8 |
| compass | 44162 | 16644 | 17 |
| wine | 44091 | 2554 | 11 |
| house_16H | 44123 | 13488 | 16 |
| MagicTelescope (Magic) | 44125 | 13376 | 10 |
| Higgs | 44129 | 940160 | 24 |
| jannis | 44131 | 57580 | 54 |
| credit | 44089 | 16714 | 10 |
| eye_movements | 44157 | 7608 | 23 |
| kddCup09_upselling (kddCup09) | 44158 | 5032 | 45 |
| road-safety | 44161 | 111762 | 32 |
| bank-marketing | 44126 | 10578 | 7 |
| phoneme | 44127 | 3172 | 5 |
| covertype | 44159 | 423680 | 54 |
| california | 44090 | 20634 | 8 |
| kdd_ipums_la_97-small (kdd_ipums_la) | 44124 | 5188 | 20 |
| MiniBooNE | 44128 | 72998 | 50 |
| pol | 44122 | 10082 | 26 |

## C Baseline details

In this section, we describe the hyperparameter search space for the baseline models. For each random split of every dataset, we find the optimal set of hyperparameters using a random sampler run for 400 trials. We utilize the Optuna library [48] for the hyperparamter tuning.

### C.1 XGBoost

For XGBoost, we adopt the hyperparameter search space used in Grinsztajn et al. [13].

Table 10: XGBoost [11] hyperparameters space.

| Parameter | Distribution |
|---|---|
| Max depth | UniformInt [1, 11] |
| Num estimators | UniformInt [100, 6100, 200] |
| Min child weight | LogUniformInt [1, 1e2] |
| Subsample | Uniform [0.5, 1] |
| Learning rate | LogUniform [1e-5, 0.7] |
| Col sample by level | Uniform [0.5, 1] |
| Col sample by tree | Uniform [0.5, 1] |
| Gamma | LogUniform [1e-8, 7] |
| Lambda | LogUniform [1, 4] |
| Alpha | LogUniform [1e-8, 1e2] |

## C.2 Multilayer perception (MLP)

For MLP, we adopt the hyperparameter search space from Grinsztajn et al. [13] and the architecture from Gorishniy et al. [31], which includes learning embeddings for categorical features. The models are trained for up to 300 epochs with early stopping, with the model that achieves the best validation score selected for evaluation. If validation scores do not improve for 40 epochs, training is stopped early. For learning rate scheduling, we use PyTorch's `ReduceOnPlateau` implementation.

Table 11: MLP [31] hyperparameters space.

| Parameter | Distribution |
|---|---|
| Num layers | UniformInt [1, 8] |
| Layer size | UniformInt [16, 1024] |
| Dropout | Uniform [0, 0.5] |
| Learning rate | LogUniform [1e-5, 1e-2] |
| Category embedding size | UniformInt [64, 512] |
| Learning rate scheduler | [True, False] |
| Batch size | [256, 512, 1024] |

## C.3 HyperFast

For HyperFast [32], we adopt the hyperparameter space from the original paper.

Table 12: HyperFast [32] hyperparameters space.

| Parameter | Distribution |
|---|---|
| N ensemble | [1, 4, 8, 16, 32] |
| Batch size | [1024, 2048] |
| NN bias | [True, False] |
| Stratify sampling | [True, False] |
| Optimization | [None, 'optimize', 'ensemble_optimize'] |
| Optimize steps | [1, 4, 8, 16, 32, 64, 128] |
| Seed | UniformInt [0, 9] |

# D Compute resources

We conducted our experiments on a variety of machines, including

- CPU: Intel(R) Xeon(R) Gold 6226R, GPU: RTX 3090
- CPU: Intel(R) Xeon(R) Gold 6426Y, GPU: RTX 4090
- CPU: Intel(R) Xeon(R) Gold 6426Y, GPU: RTX A6000

# E Comparison with CAAFE

## E.1 Main results

Table 13: **Performance improvements by OCTree on datasets with language descriptions.** We report test error rates (%) on six classification tasks (*) and mean absolute errors ($\times 10^{-3}$) for two regression tasks ($\dagger$). The lowest error is in **bold**. Values in parentheses indicate the relative error rate reduction from the baseline. We report the mean error and standard deviation across three random splits, except for the two regression tasks (time series tabular data), which are split by time index. GPT-4o was used for both CAAFE and OCTree.

| Dataset | Baseline | CAAFE [19] | OCTree (Ours) |
|---|---|---|---|
| Tesla$^\dagger$ | 6.61 | 6.36 (3.8%) | **5.48 (17.1%)** |
| Enefit$^\dagger$ | 8.00 | 7.97 (0.4%) | **7.82 ( 2.3%)** |
| Disease* | $28.09_{\pm7.9}$ | $27.46_{\pm7.6}$ (2.2%) | $\mathbf{25.72_{\pm6.6}}$ **( 8.4%)** |
| Clinical* | $46.27_{\pm5.0}$ | $45.39_{\pm4.9}$ (1.9%) | $\mathbf{43.75_{\pm4.4}}$ **( 5.4%)** |
| Academic* | $14.15_{\pm0.6}$ | $14.13_{\pm0.3}$ (0.1%) | $\mathbf{13.74_{\pm0.1}}$ **( 2.9%)** |
| BTC* | $25.11_{\pm3.7}$ | $24.67_{\pm2.9}$ (1.8%) | $\mathbf{24.00_{\pm3.1}}$ **( 4.4%)** |
| Diabetes* | $5.45_{\pm3.6}$ | $4.92_{\pm3.8}$ (9.8%) | $\mathbf{4.16_{\pm3.9}}$ **(23.6%)** |
| Student* | $36.17_{\pm2.0}$ | $36.00_{\pm2.0}$ (0.5%) | $\mathbf{35.83_{\pm2.3}}$ **( 0.9%)** |

We conduct a comparative evaluation using all datasets (along with additional ones) with contextual information from the experiments summarized in Table 1. First, CAAFE [19] exhibited high variance, even on the same dataset split, partly due to the randomness associated with GPT-4o's temperature-based sampling. At times, it failed to improve upon the baseline. To address this, we average the performance over three trials per random split and report the mean and variance. As shown in Table 13, our method consistently outperforms CAAFE. Note the official implementation of CAAFE has been slightly modified to accommodate regression tasks.

## E.2 Case study

```
f'''
df['Age Category'] = pd.cut(df['Age'], bins=[0, 30, 60, 100], labels=['Young',
↪  'Adult', 'Senior'])
df['Fever_Cough_Interaction'] = df['Fever'] * df['Cough']
'''
```

Listing 5: New columns introduced by CAAFE [19].

```
f'''
If the individual has ''Fever'' and ''Fatigue'' and ''Difficulty Breathing'' with ''Age''
↪  between 60 and 90, then predict ''Exposure to Infected Individuals'' as ''Yes''.
↪  Otherwise, predict ''Exposure to Infected Individuals'' as ''No''.
'''
```

Listing 6: New columns introduced by OCTree (Ours).

As illustrated in Listings 5 and 6, CAAFE struggles to introduce meaningful columns for the disease dataset, whereas our method proposes more relevant and coherent rules.

## F  Case studies on different types of LLMs

In Section 4.2, we evaluate three different LLMs and provide additional comparisons in this section. As shown in Table 4, our custom model (the Llama 2 model trained fine-tuned on a high-quality open dialogue dataset) achieves the lowest error rate, followed by Code Llama and the Llama 2 chat base model, both of which still outperform the baseline XGBoost. We also observed several notable differences in feature generation patterns among the models.

First of all, *Code Llama* utilizes various NumPy operations (*e.g.*, 'np.sin') to generate mathematically sophisticated features:

- $x9 = ((x4 + 0.24) * x1) + ((x5 + 0.27) * x2)$
- $x9 = np.sin(x1) * np.cos(x4)$
- $x9 = x4 * np.tan(x8)$

Secondly, *Llama 2 chat base* favors simple polynomial combinations:

- $x9 = x4 * x1 * (x2 + x6) * *2$
- $x9 = x4 * x1 * *3 * (x2 + x6) * *4 * (x7 + x8)$
- $x9 = x4 * x1 * *2 * (x2 + x6) * *3 * (x7 + x8) * (1 + x5 * *4)$

Finally, *our custom model* tends to explore a polynomial space of features while also utilizing built-in Python functions such as 'abs()'. Compared to the Llama 2 chat base model, it more effectively navigates a broader space of features:

- $x9 = x1 * *2 * (x2 - x3)$
- $x9 = abs(x1) * *x1 + x2 - x3 - x4 - x5 - x6 - x7 - x8$
- $x9 = x1 * (x1 + 2) - x2 * (x2 - 0.5) * (x2 - 0.5)$

These observations suggest that, while all three models can generate useful features, our custom model more effectively navigates a broader feature space, leading to the generation of even more valuable features. We suspect that further training on code data, particularly to enhance the ability to leverage scientific computing libraries like NumPy, could lead to additional performance improvements.

## G  Scalability of OCTree

Here, we evaluate the scalability of our method on datasets with hundreds of features (*e.g.*, 501) from the OpenML repository [49]. We choose XGBoost [11] as the baseline model, because it is the most competitive baseline in our main experiments (see Table 3). Additionally, we use GPT-3.5-Turbo as the rule generator, as the Llama 2-based model we primarily use is constrained by a relatively short context length, which becomes limiting as the prompt size in-

Table 14: **OCTree on datasets with hundreds of features.** We report the mean error (%) and the lowest error is highlighted in **bold**. Values in parentheses indicate the relative error reduction from the baseline model (*i.e.*, XGBoost [11]).

| Dataset | # features | Baseline | **OCTree (Ours)** |
|---------|-----------|----------|-------------------|
| madelon | 501 | 21.54 | **20.19 (6.3%)** |
| nomao | 119 | 3.08 | **2.84 (7.8%)** |

creases with the number of features. As shown in Table 14, our method scales effectively to datasets with a larger number of features. For example, on the madelon dataset, which contains 501 columns, our method reduces the relative error by 6.3% compared to the XGBoost baseline.

# H    Examples of rule optimization

To identify an effective feature generation rule, we conduct 50 rounds of optimization with the LLM. In each round, the LLM is provided with an optimization trajectory that includes reasoning information from previous experiments. Consequently, the input prompt evolves throughout the optimization process, with later rounds incorporating more accumulated information. Below, we show the first and last five output rules generated during the optimization on the electricity dataset.

First five:

- $\texttt{x12} = \texttt{x3} + (\texttt{x5} * \texttt{x8}) * \texttt{x2} - \texttt{x6}$
- $\texttt{x12} = \texttt{x5} * \texttt{x2} + (\texttt{x3} * \texttt{x4} - \texttt{x6}) - \texttt{x1}$
- $\texttt{x12} = \texttt{x11} + (\texttt{x7} * \texttt{x10}) * \texttt{x2} - \texttt{x8}$
- $\texttt{x12} = \texttt{x3} + (\texttt{x6} * \texttt{x8} * \texttt{x2}) - \texttt{x1}$
- $\texttt{x12} = (\texttt{x1} * \texttt{x6} + \texttt{x2} * \texttt{x8}) * \texttt{x3} - \texttt{x7}$

Last five:

- $\texttt{x12} = ((\texttt{x2} - \texttt{x1}) * (\texttt{x11} - \texttt{x4})) * *2 - (0.5 * (\texttt{x1} - 0.3))$
- $\texttt{x12} = ((\texttt{x2} - \texttt{x1}) * (\texttt{x11} - \texttt{x4})) * *6$
- $\texttt{x12} = ((\texttt{x2} - \texttt{x1}) * (\texttt{x11} - \texttt{x4})) * *14 - (0.02 * (\texttt{x5} - \texttt{x7})) - 0.5$
- $\texttt{x12} = (\texttt{x11} - \texttt{x1}) + (\texttt{x11} * (\texttt{x6} - \texttt{x7}))$
- $\texttt{x12} = ((\texttt{x2} - \texttt{x1}) * (\texttt{x11} - \texttt{x4})) * *3 + (0.1 * (\texttt{x5} - \texttt{x6})) * *2$

Note that in the early stages of optimization, the LLM tends to explore a wider variety of rules, allowing for greater exploration. In contrast, during the later stages, the model narrows its focus to refine features within a more constrained space.

# I    Broader impacts

Our method is particularly effective in scenarios where collecting real data is costly or restricted, such as in the finance or medical domains, where data availability is often limited due to privacy concerns. However, since the features generated by OCTree are artificial, it is crucial to carefully inspect these features for their relevance and reliability.

