# OpenReview forum: "Optimized Feature Generation for Tabular Data via LLMs with Decision Tree Reasoning"
_NeurIPS.cc/2024/Conference — NeurIPS 2024 poster_

### Official Review · Reviewer_8DXj · 2024-07-04

**Soundness:** 2
**Presentation:** 3
**Contribution:** 2
**Rating:** 4
**Confidence:** 4

**Summary:**

The paper introduces OCTree, a framework that uses LLMs to generate and refine feature generation rules for tabular data. By incorporating decision tree reasoning, OCTree iteratively improves feature generation using feedback from past experiments. The framework enhances the performance of some prediction models, including decision trees and neural networks, and demonstrates improvements on real-world datasets. OCTree works with datasets with or without language descriptions and outperforms existing feature engineering methods, showing flexibility in feature generation for prediction tasks.

**Strengths:**

The paper combines LLMs with decision tree reasoning to generate features for tabular data, offering a novel approach to feature generation. The paper demonstrates improvements through several experiments, and its flexible framework shows potential benefits in enhancing model performance. The paper is well-structured and clearly written.

**Weaknesses:**

1. The framework is costly and time-consuming due to the need for training new models for validation during each iteration, and it involves some manual operations, preventing full automation.
2. Lack of comparisons of training times makes it difficult to assess the efficiency relative to other approaches.
3. The experiments are limited to datasets with a maximum of 54 features, lacking evidence of effectiveness on high-dimensional data (e.g., datasets with hundreds of features).
4. Datasets are sampled to sizes smaller than 50,000 instances, raising concerns about effectiveness and scalability on larger datasets.

**Questions:**

1. Are there other works that use LLMs for feature generation? If so, how does this approach differ from them?
2. What is the time cost associated with OCTree? How much slower is it compared to directly training XGBoost? Does the time cost increase significantly with the size of the dataset and the feature dimensionality?

---

> ### Author Rebuttal · Authors · 2024-08-07
>
> Dear Reviewer 8DXj,
>
> We sincerely appreciate the time and effort you dedicated to reviewing our manuscript and providing insightful comments. Below, we address each of your points individually.
>
> ---
> **[W1, 2] The framework is costly and time-consuming (with no training time comparison) and involves some manual operations.**
>
> Thank you for your comment. As we described in Section 5 of our manuscript, one potential limitation is that it may be time-consuming if the prediction model requires extensive training. In this regard, we acknowledge that our method could be slower compared to competing methods like OpenFE [3]. Thus, we focused on evaluating the effectiveness of our method rather than the run time. However, once the feature generation rules are optimized, new column features can be generated simply by applying these rules to any additional data that becomes available.
>
> As we also described in our draft, there are several approaches to further scaling our method:
> - Feature transfer: As demonstrated in Table 6, it is possible to first generate features using a simpler model like XGBoost, and then transfer these features to the more complex target model like MLP. This allows for faster rule generation for more complex prediction models.
> - Use of open LLMs: Unlike methods that rely on proprietary LLMs such as ChatGPT, we demonstrated that open models like Llama 2, with minimal additional fine-tuning, can be highly effective in our feature generation framework. This approach avoids the cost associated with paid APIs.
> - Use of data subsets: We have shown our method is effective across datasets of varying sizes. This suggests that during rule generation, a suitable subset of the training data can be used to accelerate optimization, enabling scalability to larger datasets.
>
> Lastly, we would like to point out that the only significant manual task involved is designing an appropriate prompt for the rule generator LLM. The version used in our experiments will be fully released alongside the code.
>
> ---
> **[W3] Scalability: Number of features.**
>
> Thanks for the valuable suggestion. In response, we evaluate the scalability of our method on datasets with hundreds of features (e.g., 501) using datasets from the OpenML repository. We chose XGBoost as the baseline because it was the most competitive baseline in our main experiments (see Table 2). Additionally, we used GPT-3.5-Turbo as the rule generator because the Llama2-based model we primarily used in our work is constrained by a maximum context length of 2048, which becomes limiting as the prompt size increases with the number of features. However, we emphasize that our method is compatible with an arbitrary LLM as the rule generator, as shown in Tables 1 and 3 of our manuscript.
>
> As shown in the table below, our method scales effectively to datasets with a larger number of features. For example, on the madelon dataset, which contains 501 columns, our method reduces the relative error by 6.3% compared to the XGBoost. Here, we report test error, with values in parentheses indicating the reduction in relative error rates. We will incorporate these results into the final draft.
>
> \begin{array}{lccc}
> \hline
> \text{Dataset}&\text{\\# features}&\text{Baseline}&\textbf{OCTree (Ours)}\newline\hline
> \text{madelon}&501&21.54&\textbf{20.19 (6.3\\%)}\newline
> \text{nomao} &119&3.08&\textbf{2.84 (7.8\\%)}\newline\hline
> \end{array}
>
> ---
> **[W4] Scalability: Number of samples.**
>
> Thanks for the valuable suggestion. Following the suggestion, we evaluate the scalability of our method on larger datasets from the OpenML repository using XGBoost. As shown in the table below, our method scales effectively to datasets of a much larger scale. For example, in the nyc-taxi-green-dec-2016 dataset, which contains over 500,000 samples, our method achieves a 13.7% reduction in relative error compared to the baseline. Here, we report test error, with values in parentheses indicating the relative error reductions. We will incorporate these results into the final draft.
>
> \begin{array}{lccc}
> \hline
> \text{Dataset}&\text{\\# samples}&\text{Baseline}&\textbf{OCTree (Ours)}\newline\hline
> \text{nyc-taxi-green-dec-2016}&\text{581,835}&2.91&\textbf{2.51 (13.7\\%)}\newline
> \text{Covertype}&\text{423,680}&2.79&\textbf{2.15 (22.9\\%)}\newline\hline
> \end{array}
>
> ---
> **[Q1] Are there other works that use LLMs for feature generation? How does this approach differ from them?**
>
> As described in Section 2 of our manuscript, CAAFE [1] proposes a context-aware automatic feature engineering method that leverages LLMs but is distinct from OCTree in several key ways. Please refer to the global response above for more details. We thank the reviewer for the question and will incorporate the results in the final manuscript.
>
> ---
> **[Q2] Associated time cost.**
>
> Thanks for your question regarding various aspects of our method's efficiency.
>
> First, the primary time cost comes from (a) using the LLM to suggest rules and (b) training the prediction model on the suggested features to compute validation scores. Directly training an XGBoost would involve a single training run on the original dataset, whereas our method involves multiple iterations until the optimal features are generated. However, once the rules have been generated, applying these rules to any additional data is trivial.
>
> Second, while dataset size and feature dimensionality mainly affect the training time of the prediction model, they have minimal effect on LLM inference. Additional features may slightly increase the prompt length, but the overall size of the training dataset has little impact on inference time.
>
> Lastly, please refer to the approaches to scaling the method we outlined in our response to [W1, 2]. These include using simpler prediction models and leveraging feature transfer, using open LLMs of moderate size like Llama 2 at the 7B scale, and using a subset of the training data during rule generation to improve efficiency.

---

> > ### Comment · Reviewer_8DXj · 2024-08-13
> >
> > I appreciate the author's efforts in addressing the questions and concerns. However, there are still some critical issues that remain unresolved.
> >
> > 1. For [W1, 2], although this paper focuses on the method's effectiveness, the improvement over the baseline in many datasets is not substantial (for example, in Table 2, where the comparison with XGBoost shows that in 19 datasets, the relative error rate reduction is less than 4% in 11 datasets). Therefore, I believe that a training time comparison is necessary.
> > 2. For [W4], the main issue is not whether the method performs well on one or two datasets with more than 50,000 instances, but rather that the paper's experimental section should avoid sampling datasets down to fewer than 50,000 instances, as this is likely to cause the baselines to underperform.
> > 3. For [Q1], since there are related comparative methods, a comparison between this method and those methods should be included in all evaluation parts of the paper, e.g., Table 2.

---

> > > ### Author Response · Authors · 2024-08-13
> > > **Thank you for the additional feedback**
> > >
> > > Dear Reviewer 8DXj,
> > >
> > > Thank you for the additional feedback. Here, we would like to further address your concerns individually.
> > >
> > > ---
> > > **For [W1, 2], although this paper focuses on the method's effectiveness, the improvement over the baseline in many datasets is not substantial (for example, in Table 2, where the comparison with XGBoost shows that in 19 datasets, the relative error rate reduction is less than 4% in 11 datasets). Therefore, I believe that a training time comparison is necessary.**
> > >
> > > Thank you for your feedback. We understand the importance of evaluating training time. The run time of our method depends on the number of additional features generated and the number of optimization steps used to refine each feature. As you noted, these factors can be substantial depending on dataset size and model type.
> > >
> > > Nevertheless, we would like to highlight that our method demonstrates consistent improvement in performance across (i) datasets and (ii) baseline model types. As shown in Table 4 of our manuscript, our method outperforms competing automatic feature engineering methods, including AutoFeat and OpenFE. In particular, OpenFE fails to introduce useful features for multilayer perceptron (MLP) in 9 of the 22 classification datasets used in Tables 1 and 2, which contrasts with our method's more robust performance.
> > >
> > > In response to your comment, we will assess potential speed-ups through approaches such as feature transfer and reducing training data during feature generation that we highlighted in our previous response. We will include these evaluations and additional quantitative results on training time in the final revision.
> > >
> > > ---
> > > **For [W4], the main issue is not whether the method performs well on one or two datasets with more than 50,000 instances, but rather that the paper's experimental section should avoid sampling datasets down to fewer than 50,000 instances, as this is likely to cause the baselines to underperform.**
> > >
> > > Thank you for highlighting this concern. We sampled datasets down to fewer than 50,000 instances to align with the experimental setup used in [1] (see Appendix A.2.2), which provides a comprehensive benchmarking study of tree-based and deep learning methods on tabular data. This alignment facilitates a clearer comparison of our results with those from previous works.
> > >
> > > We also want to clarify that once a dataset is sampled, the same sampled dataset is used for both the baselines and our method. Thus, while sampling may impact baseline performance, as you pointed out, our feature generation method is subject to the same conditions. Additionally, it's relevant to note the experimental setup of the previous automatic feature engineering method, CAAFE [2], which only focuses on small datasets with up to 2,000 samples.
> > >
> > > In contrast, our method has been evaluated on datasets with significantly larger feature dimensions and sample sizes, as detailed in the additional experiments provided in our previous response. We hope these results address your original concern and we will ensure that these results are clearly incorporated into the final revision to comprehensively address your concerns.
> > >
> > > [1] Grinsztajn et al., Why do tree-based models still outperform deep learning on tabular data? NeurIPS 2022.\
> > > [2] Hollman et al., LLMs for Semi-Automated Data Science: Introducing CAAFE for Context-Aware Automated Feature Engineering, NeurIPS 2023.
> > >
> > > ---
> > > **For [Q1], since there are related comparative methods, a comparison between this method and those methods should be included in all evaluation parts of the paper, e.g., Table 2.**
> > >
> > > Thank you for your feedback. We would like to clarify that we have compared our approach to CAAFE on **all** datasets where CAAFE is applicable, as detailed in our **global response** and the **attached PDF**. Specifically, we performed a comparative evaluation using all datasets that include contextual information, as shown in Table 1 of our manuscript. Since CAAFE requires language-based contexts, which are not available for all datasets (e.g., those used in Table 2), our comparison focused on datasets where CAAFE can be effectively applied. This highlights the broader applicability of our method compared to CAAFE. We hope this explanation clarifies our approach and the scope of our comparisons.

---

### Official Review · Reviewer_245r · 2024-07-10

**Soundness:** 2
**Presentation:** 1
**Contribution:** 1
**Rating:** 3
**Confidence:** 3

**Summary:**

The authors propose an automatic feature engineering method called OCTree. The OCTree algorithm uses LLMs to generate new features. The feature is then used in the training of the black box, and the new validation score is stored. The algorithm uses a decision tree to learn rules that show the reasoning behind selecting these new features and incorporate this in an iterative process.


NOTE: If the authors address the issues I have raised in limitations and answer my questions, I will improve my score.

**Strengths:**

The paper's research question, automatic feature engineering, is an important area of machine learning research that has not gained enough attention.

**Weaknesses:**

It is very difficult for me to follow the notation and understand what the algorithm is doing. This gets harder as the authors have not released the code for the experiments.

The authors keep iterating that the main limitation of another feature engineering is that they require setting the manual search space without showing how their approach is solving this limitation.

The experiments do not include CAAFE, the most relevant algorithm to OCTree. The authors claim that the reason behind this is that CAAFE relies on a language-based context of the dataset, making it inapplicable to the datasets they have used. But I am not yet convinced.

None of the decision-tree reasoning and how it is improving OCTree is explained in the paper.

**Questions:**

Why have the authors excluded CAAFE? I looked at the tutorials, and they seem very easy to use.

Why do the authors say that CAAFE is orthogonal to their approach? I do not think that is the case. I understand that your approach is CAAFE + a decision tree process embedded. Can you elaborate on the difference between your approach and CAAFE with a concrete example?

Your notations are so confusing. In Line 131, what does D + r mean? D is a dataset, and r is a rule; they cannot be summed up.

Can authors present evidence on how the inclusion of rules learned by decision trees is increasing the efficiency of their algorithm?

**Limitations:**

The main competitive algorithm in this work, CAAFE, is excluded from the study. Therefore, we do not have a good baseline for how this approach can improve the problem.

The authors have not addressed the study's main concern: the risk of memorization. LLMs are possibly trained on the open datasets they are using. This could be that the LLM is, for example, summarizing what most Kaggle blogs and Medium posts on these datasets have done.

---

> ### Author Rebuttal · Authors · 2024-08-07
>
> Dear Reviewer 245r,
>
> We sincerely appreciate the time and effort you dedicated to reviewing our manuscript and providing insightful comments. Below, we address each of your points individually.
>
> ---
> **[W1] Difficulty following the notation and the algorithm. Authors should release the code.**
>
> Thank you for your feedback. While several reviewers have noted that the paper is generally well-structured and clearly written, we acknowledge that some notations could be clearer. We will review and clarify these notations to improve understanding.
>
> Regarding the availability of our code, we plan to release it in full. However, due to anonymity requirements, we are providing a version with any author-identifiable components removed. Per the guidelines, we have submitted the link to the AC, so that it can be made available for your review.
>
> ---
> **[W2] How does the proposed method tackle the limitations of other feature engineering methods that require setting the manual search space?**
>
> We would like to clarify that our approach enables the LLM to automatically generate rules without requiring a predefined optimization space. In contrast, other feature engineering methods, such as OpenFE [3], depend on predefined search spaces (e.g., using predefined operators like + or -) to generate candidates. For example, we use prompts like “Give me a new rule that is different from the old ones and has a score as high as possible,” therefore generating features with the rules we get from LLM’s suggestions alone, without any specification of the search space (see Section A.3 of our manuscript). Consequently, our LLM-based framework fully automates the optimization process without the need for a predefined search space.
>
> ---
> **[W3, Q1, Q2, L1] Comparison with CAAFE [1] and why orthogonal?**
>
> We would first like to clarify that, as mentioned in Section 4.1 of our manuscript, we originally excluded CAAFE from our comparison as it requires a language-based context for the dataset. This requirement limits CAAFE's applicability in cases where language descriptions are not explicitly provided, such as when feature names and values are obfuscated to protect confidentiality, as is common in financial and medical datasets. In contrast, the methods that we evaluated in Table 4, as well as our approach, are context-agnostic and can be applied to datasets without such linguistic descriptions. However, our method does benefit from clear language descriptions, as shown in Table 1, when available.
>
> To further address your concern, we have compared our method to CAAFE and highlighted some key differences and distinctions. Please refer to the global response above for more details. We thank the reviewer for the suggestion and will incorporate the results in the final manuscript.
>
> ---
> **[W4, Q4] Decision tree reasoning: How it improves OCTree is not explained in the paper and needs evidence on how including rules increases the algorithm’s efficiency.**
>
> We would like to first clarify that decision tree reasoning refers to the language description of a decision tree constructed from the entire dataset, including any newly generated features. We have already conducted an ablation study, detailed in Table 5, to assess the effects of providing decision tree reasoning as feedback to the rule generator LLM. Here, our findings indicate that incorporating decision tree reasoning leads to the generation of higher-quality features, resulting in an additional performance improvement of up to 5%. This is because decision tree reasoning provides valuable insights learned from the entire training set, as it highlights the columns that are considered more significant (as nodes in the tree) and the corresponding values (as thresholds of the nodes) used for prediction. We will incorporate these points more clearly into our final manuscript.
>
> ---
> **[Q3] Clarification of notation: Meaning of $D \oplus r$.**
>
> We would like to clarify that the notation $D \oplus r := \\{x_i \oplus r(x_i), y_i \\}_{i=1}^{N}$ represents the dataset in which each sample $x_i$ is augmented with an additional feature generated by applying the rule $r$. To address your concerns and enhance clarity, we plan to update this notation to $D^r$ in the final draft.
>
> ---
> **[L2] Risk of memorization.**
>
> We would like to emphasize that our method is completely free from the risk of memorization because it generates new column features that are not present in the original dataset. Even if the LLM has memorized information from open datasets, our approach does not rely on the specific data; instead, it generates and introduces entirely new columns to the dataset.
>
> To further address your concern regarding the potential impact of the LLM being trained on open datasets, we would like to highlight that most of the datasets used in Table 1 (e.g., Tesla, Enefit, Disease, Clinical) were released after the release of Llama 2, our base LLM. The fact that our method enhances the performance of various baseline models on this dataset demonstrates its capability to generate useful features without relying on dataset-specific information. Moreover, Table 2 presents the results of experiments conducted on datasets that lack semantic information about the columns. We used generic indicators (e.g., 'x1', 'x2') for columns and applied ordinal encoding and a min-max scaler to transform all feature values. Even in this setting, our method was able to enhance the performance of a range of baseline models across datasets of varying sizes and characteristics.

---

> > ### Comment · Reviewer_245r · 2024-08-11
> >
> > I thank the authors for answering my concerns. I appreciate it. Unfortunately, I am not convinced after their answers.
> >
> > Some more feedback is below:
> >
> > > While several reviewers have noted that the paper is generally well-structured and written
> >
> > Absolutely. That's the point of peer review. However, I would like to say that writing does not necessarily mean that the notations are accurate. It can be seen that the "story" is well written.
> >
> > > This requirement limits CAAFE's applicability in cases where language descriptions are not explicitly provided, such as when feature names and values are obfuscated to protect confidentiality, as is common in financial and medical datasets.
> >
> > Sure, but then you must also compare CAAFE to your approach in cases where it is possible. Or is your approach ONLY applicable to those cases? In that case, I would say your solution is very niche.
> >
> > >We have already conducted an ablation study, detailed in Table 5, to assess the effects of providing decision tree reasoning as feedback to the rule generator LLM.
> >
> > I appreciate it, but what I mean is that we need to see the decision tree rules and how the tree relates to the process.
> >
> > >Even if the LLM has memorized information from open datasets, our approach does not rely on the specific data; instead, it generates and introduces entirely new columns to the dataset.
> >
> > Yeah, but you cannot really say that those features do not come from a tutorial on feature engineering on Kaggle in 2015, right? That is what I meant.

---

> > > ### Author Response · Authors · 2024-08-12
> > > **Thank you for the additional feedback (1/2)**
> > >
> > > **[F1] Absolutely. That's the point of peer review. However, I would like to say that writing does not necessarily mean that the notations are accurate. It can be seen that the "story" is well written.**
> > >
> > > We appreciate the constructive feedback and agree that clarity in notation is crucial. As we noted in our earlier response, we recognize that certain notations (e.g., $D \oplus r$) could be made clearer. We will carefully review and refine these notations in the final draft to ensure ease of understanding.
> > >
> > > ---
> > > **[F2] Sure, but then you must also compare CAAFE to your approach in cases where it is possible. Or is your approach ONLY applicable to those cases? In that case, I would say your solution is very niche.**
> > >
> > > We would like to clarify that our method is applicable to both cases where language descriptions are clearly provided and where they are not (see Tables 1 and 2 in our manuscript). We fully agree that comparing our approach with CAAFE is important. Therefore, as per your suggestion, we have already compared CAAFE to our approach in the previous rebuttal using all datasets with contextual information from the experiments summarized in Table 1 of our manuscript.
> > >
> > > To summarize the results, Table 1 of the attached PDF shows that our method consistently outperforms CAAFE. For a more detailed discussion, we kindly refer you to the **global response** and the **attached PDF**. We appreciate your suggestion and will incorporate these comparative results in the final draft.
> > >
> > > ---
> > > **[F3] I appreciate it, but what I mean is that we need to see the decision tree rules and how the tree relates to the process.**
> > >
> > > Thank you for your question and the opportunity to provide further clarification. First of all, we would like to clarify that decision tree reasoning refers to the language description of a decision tree that is constructed from the entire dataset, including any newly generated features. This reasoning is then used to guide the LLM in generating more effective feature generation rules.
> > >
> > > For instance, consider the following example:
> > > - Original columns: Fever, Breathing
> > > - Introduced column: Smoking status
> > > - Task: Predicting whether a patient has a disease
> > >
> > > A decision tree derived from this data might produce reasoning such as:
> > >
> > > ```
> > > if ‘Has difficulty breathing’:
> > >      if ‘Has fever’:
> > >           ‘Subject has a disease’
> > >      else:
> > >           ‘No disease’
> > > …
> > > ```
> > >
> > > This decision tree reasoning is provided to the LLM as feedback to refine its feature generation process. Here, we provide a prompt that is used to guide LLM to find good feature generation rules (see Appendix A.3).
> > >
> > > ```
> > > I have some rules to predict {Objective} with attributes listed below.
> > > - {Column #1 Name} (Numerical value of {Column #1 Min} ~ {Column #1 Max}):  {Column #1 Feature Description}
> > > - {Column #2 Name} (Boolean):  {Column #2 Feature Description}
> > > - {Column #3 Name} (Categorical value of {Value #1}, {Value #2}, {Value #3}):  {Column #3 Feature Description}
> > > ...
> > >
> > > We also have corresponding decision trees (CART) to predict {Objective} from the attributes listed above along with predicted {New Column Name}.
> > > The rules are arranged in ascending order based on their scores evaluated with XGBoost classifier, where higher scores indicate better quality.
> > >
> > > Rule to predict {Objective}:
> > > {Rule #1}
> > > Decision tree (CART):
> > > {Decision Tree #1}
> > > Score evaluated with XGBoost classifier:
> > > {Score #1}
> > >
> > > Rule to predict {Objective}:
> > > {Rule #2}
> > > Decision tree (CART):
> > > {Decision Tree #2}
> > > Score evaluated with XGBoost classifier:
> > > {Score #2}
> > >
> > > ...
> > >
> > > Give me a new rule to predict {Objective} that is different from the old ones (but should use the listed attributes above) and has a score as high as possible.
> > >
> > > Improved rule:
> > > ```
> > > We appreciate the question and will provide more examples and explanations in the final draft.

---

> > > > ### Author Response · Authors · 2024-08-12
> > > > **Thank you for the additional feedback (2/2)**
> > > >
> > > > **[F4] Yeah, but you cannot really say that those features do not come from a tutorial on feature engineering on Kaggle in 2015, right? That is what I meant.**
> > > >
> > > > Thank you for your insightful comments. We understand your concern that LLMs may have memorized feature engineering techniques, such as those from the tutorials on Kaggle. Indeed, our method leverages LLM’s ability to introduce engineering rules from existing features. However, while LLMs may be familiar with established techniques, this does not guarantee that they can suggest effective feature engineering rules for every dataset. For example, without contextual information about a column, an LLM may struggle to identify which features are most relevant and, therefore, may not provide useful rules.
> > > >
> > > > Our method addresses this limitation by proposing an optimization process to find good feature generation rules by designing novel and effective feedback mechanisms. Specifically, in the early optimization stages, the LLM explores a wide range of potential rules, which may be based on common engineering techniques learned, as you pointed out. As the process continues, the LLM refines the solution space around previously discovered solutions, making only minor adjustments during the later stages. In short, our method goes beyond simply utilizing the LLM’s existing knowledge of feature engineering techniques by actively guiding it toward discovering the most effective rules for the specific dataset.
> > > >
> > > > We appreciate your feedback and will incorporate this discussion into the final draft.

---

### Official Review · Reviewer_Pmbp · 2024-07-12

**Soundness:** 2
**Presentation:** 3
**Contribution:** 3
**Rating:** 6
**Confidence:** 5

**Summary:**

This paper proposes a new tabular learning framework called OCTree (Optimizing Column Feature Generator with Decision Tree Reasoning). The framework leverages the reasoning capabilities of large language models (LLMs) to automatically generate new column features based on feedback from decision trees. Experimental results demonstrate that this framework consistently outperforms existing automatic feature engineering methods across various tabular data prediction tasks.
The OCTree framework effectively utilizes the reasoning capabilities of LLMs and feedback from decision trees to automatically generate new column features, significantly enhancing the performance of tabular data prediction tasks. This framework demonstrates the great potential of LLMs in automatic feature engineering for tabular data.

**Strengths:**

S1: A new framework for automatic feature generation that leverages LLMs' language understanding and reasoning capabilities, using feedback from decision trees.

S2: All experiments on various real-world datasets show significant performance improvements in different prediction models, including gradient-boosted decision trees and deep neural networks.

S3: The method works for both datasets with and without language descriptions and shows good transferability of generated features across different types of models.

**Weaknesses:**

w1: I thoroughly enjoyed reading this work. The authors present a unique perspective on the optimization of tabular data tasks. For example, in the abstract, the authors highlight the shortcomings of existing work, such as "they often rely solely on validation scores to select good features, neglecting valuable feedback from past experiments that could inform the planning of future experiments." This indeed reflects the current issues faced by Automated Feature Transformation, where high-performing features may not necessarily be meaningful features. However, conversely, it seems that the goal of Feature Engineering is to improve the performance of downstream machine learning tasks, which is a very direct objective. Isn't it contrary to the original intent of feature engineering if validation scores are not used as an evaluation metric?

w2: The writing of the entire paper is relatively easy to read, although some concepts were not discussed in depth. For instance, the authors only claim that validation scores are a flawed feedback mechanism but do not explore some of the newer methods in the main text, such as the latest approaches utilizing reinforcement learning [1-2] or generative artificial intelligence [3-4]. These methods employ the performance of models like decision trees and other machine learning models as scores to learn strategies.

w3: In Figure 1, Fever, Fatigue, Breathing, etc., are used through processes like Tabular Data Understanding and rule generation to combine boolean forms (Fever, Fatigue) or one-hot forms of Breathing to create a new feature, Smoke. This is quite intuitive; however, if the rules generated in step 1 are inaccurate, the generated column might also be inaccurate. Moreover, can the concept of whether one smokes or not be automatically and adaptively formed during the generation of decision trees or the training of neural networks, rather than requiring such an inaccurate combination? If not, do the authors have experiments or specific mathematical proofs to validate this claim or to show that neural networks, for instance, would converge more difficultly without these generated key features?

w4: In the Column name generation section of the Methodology (section 3.2), how do the authors ensure that the generated column names are valid?

w5: In Table 2, it seems that some datasets with non-linguistic descriptions did not improve or only showed general improvement. What are the specific reasons for this?

Although this work seems overly intuitive at times, I remain excited about the application of large language models in this field. I hope the authors can address the above questions.

[1] Traceable group-wise self-optimizing feature transformation learning: A dual optimization perspective, TKDD
[2] Group-wise Reinforcement Feature Generation for Optimal and Explainable Representation Space Reconstruction, KDD
[3] Reinforcement-Enhanced Autoregressive Feature Transformation: Gradient-steered Search in Continuous Space for Postfix Expressions, NeurIPS
[4] DIFER: Differentiable Automated Feature Engineering, ICAML

**Questions:**

see weakness

**Limitations:**

see weakness

---

> ### Author Rebuttal · Authors · 2024-08-07
>
> Dear Reviewer Pmbp,
>
> We sincerely appreciate the time and effort you dedicated to reviewing our manuscript and providing insightful comments. Below, we address each of your points individually.
>
> ---
> **[W1] Isn’t it contrary to the original intent of feature engineering if validation scores are not used as an evaluation metric?**
>
> We would like to first clarify that in our framework, validation scores are indeed a key objective for optimization and are provided as feedback to the rule generator LLM. However, instead of providing just the most recent validation score, we additionally provide a history of the scores to help the LLM better understand the optimization landscape and iteratively refine and improve the rules it generates. We also found that incorporating decision tree reasoning as input is beneficial (see Table 5 in our manuscript). This reasoning serves as a summary of the training dataset, giving the LLM a deeper understanding of the data's structure and relationships. In summary, utilizing information such as a history of validation scores and insights into the training data is essential for enabling the rule generator LLM to effectively handle this complex optimization task.
>
> ---
> **[W2] In-depth discussion: The authors only claim that validation scores are flawed feedback mechanisms but need to explore some of the newer methods in the main text.**
>
> Thanks for your feedback and for pointing out the related work. As we highlighted above, we want to clarify that validation scores are indeed a key objective we optimize for in our framework. However, we have empirically shown that relying solely on validation scores in a greedy manner can be suboptimal. Our findings show that it is crucial to consider sufficient history of the scores to effectively navigate the optimization landscape. Additionally, incorporating inputs like decision tree reasoning can improve feature quality by providing a richer context for the task.
>
> Regarding related methods, framing feature generation as a sequential decision-making problem allows techniques such as reinforcement learning to be applied. Similar to our basic approach, these methods generate feature candidates and evaluate them on downstream tasks for iterative improvement. However, some of these methods require solving multiple Markov decision processes [4, 5], which can be complex and computationally demanding. In contrast, our framework offers a conceptually much simpler approach that is easier to implement. Our experiments show that an open LLM of moderate size can effectively serve as a rule generator across various datasets and prediction models. With optional fine-tuning to enhance chat or code generation capabilities, the model performs well without needing additional training or customization for each specific setting.
>
> We will incorporate these points more clearly into our manuscript.
>
> ---
> **[W3-1] If the rules generated in step 1 are inaccurate, the generated column might also be inaccurate.**
>
> While it is true that the rule generator LLM can occasionally suggest suboptimal rules, we provide feedback on previously generated rules, guiding the LLM to iteratively refine and enhance its rule generation. This feedback loop is an important component of our framework, allowing the LLM to converge toward more effective and accurate rules.
>
> ---
> **[W3-2] Can the concept automatically be formed during the generation of decision trees or the training of neural networks rather than requiring such an inaccurate combination?**
>
> You are indeed correct that decision trees inherently represent a disjunction of conjunctions of feature values and that neural networks learn latent features through hidden layers. However, our empirical findings suggest that generating explicit features using our approach and incorporating them as additional inputs yields better results than relying solely on these models to infer implicit features. We suspect this is because providing relevant features as explicit inputs enables these models to allocate their capacity more effectively, focusing their degrees of freedom on learning more sophisticated combinations of features that capture subtle patterns within the data.
>
> ---
> **[W3-3] If not, do the authors have experiments to validate this claim?**
>
> We have indeed conducted empirical evaluations to examine the impact of our generated features on model performance. Specifically, we compared the performance of XGBoost and MLP models trained without additional features (relying solely on the implicit features learned by the models) against models trained with the features generated using our framework as explicit input.
>
> As shown in Tables 1 and 2 in our manuscript, we have observed performance improvements on most datasets, with some showing relative performance gains of more than 20%. These results show that providing the generated features as explicit input can significantly improve the model's ability to converge and perform well, compared to relying solely on implicit feature learning.
>
> ---
> **[W4] Ensuring generated column names are valid.**
>
> We would like to clarify that we have already verified that the LLM indeed generates valid column names. Specifically, we have confirmed that (i) it is beneficial to use the actual values of the generated columns if they are available, and (ii) when given a candidate for a new column name, the LLM can effectively distinguish between columns that are more relevant to the target task (see Section 4.2 of our manuscript).
>
> ---
> **[W5] Table 2: Some datasets did not improve or only showed general improvement. Why?**
>
> We hypothesize that the limited effectiveness observed with some datasets with non-linguistic descriptions stems from the absence of meaningful semantic information that the LLM could leverage during rule generation. However, we would like to emphasize that for most datasets, the generated features enhanced the performance of the prediction models evaluated.

---

> > ### Comment · Reviewer_Pmbp · 2024-08-10
> >
> > I appreciate the authors' effort in addressing my questions in their response. As reflected in my scores，my overall assessment is positive, and I maintain my current scores.

---

> > > ### Author Response · Authors · 2024-08-11
> > > **Thank you very much for the response**
> > >
> > > Dear reviewer Pmbp,
> > >
> > > Thank you very much for letting us know! We are happy to hear that our rebuttal addressed your questions well.\
> > > Also we thank you for your prompt response.\
> > > If you have any further questions or suggestions, please do not hesitate to let us know.
> > >
> > > Thank you very much,\
> > > Authors

---

### Official Review · Reviewer_6JM8 · 2024-07-13

**Soundness:** 3
**Presentation:** 3
**Contribution:** 3
**Rating:** 6
**Confidence:** 4

**Summary:**

The authors introduce a novel feature engineering technique that leverages LLMs for language-based reasoning and considers the outcomes of past experiments as feedback for iterative rule improvements.

**Strengths:**

- the authors introduce a novel feature engineering technique that leverages LLMs so that past experience can be considered when proposing novel features.
 - the paper is well-structured, the results are clearly presented, and multiple LLMs are assessed

**Weaknesses:**

- the authors did not compare their method against other techniques for automated feature generation
- the authors provide no details on whether several rounds of LLM prompting were executed, how do results vary across multiple executions and whether they had issues with LLM hallucinations

**Questions:**

We consider the paper interesting and relevant. Nevertheless, we would like to point to the following improvement opportunities:

GENERAL COMMENTS

(1) - The authors compare their model against models on which such feature generation was not applied, showing promising results. Nevertheless, we miss a comparison against similar methods that generate features either with heuristics or LLMs, as explained in the Related Work section. In particular, we would encourage the authors to include some comparison against the following work: Hollmann, Noah, Samuel Müller, and Frank Hutter. "Large language models for automated data science: Introducing caafe for context-aware automated feature engineering." Advances in Neural Information Processing Systems 36 (2024).

(2) - Did the authors execute several rounds of feature generation using the same prompt and input? Did they observe any variability regarding the results obtained?

(3) - How are results different when considering different LLMs? Are there certain patterns regarding the features generated/not generated? Furthermore, the authors mention that "incorporating code datasets alongside dialogue datasets may further enhance the performance of the rule generator LLM". How were such prompt results better from regular LLMs and different from those generated by the LLM proposed by the authors (e.g., in Table 3)?

(4) - Did the authors face LLM hallucinations? We would appreciate a brief description on how did they handle such cases and some quantification, to understand how frequent such problem is across different LLMs they tried.


FIGURES

(6) - Figure 3: (i) can be made monochromatic, (ii) are datasets balanced? How is the cut threshold determined to measure Accuracy? Could this be replaced with a threshold-independent metric?

TABLES

(7) - Table 7: bold the best results

**Limitations:**

The authors acknowledge some limitations of their work. Nevertheless, we consider some additional weaknesses (see weaknesses section) should be addressed and eventually acknowledged.

---

> ### Author Rebuttal · Authors · 2024-08-07
>
> Dear Reviewer 6JM8,
>
> We sincerely appreciate the time and effort you dedicated to reviewing our manuscript and providing insightful comments. Below, we address each of your points individually.
>
> ---
> **[W1, Q1] The authors did not compare their method to other feature generation techniques, in particular, CAAFE [1].**
>
> We want to first clarify that we have already compared our method with other techniques for automated feature generation, such as AutoFeat [2] and OpenFE [3], and verified that our method performs considerably better (see Table 4 in our manuscript). Furthermore, unlike competing methods, our approach generates semantically meaningful features that human annotators can use to provide additional labels, which can significantly enhance model performance. For example, as shown in Figure 3, collecting real values for the suggested features significantly improves patient mortality prediction.
>
> To address your concern further, we compared our method to CAAFE. Please refer to the global response above for additional details. We thank the reviewer for the suggestion and will incorporate the results in the final manuscript.
>
> ---
> **[W2-1, Q2] Number of prompting rounds and results across multiple executions.**
>
> To find an effective feature generation rule, we execute 50 rounds of LLM prompting. In each round, LLM is provided with an optimization trajectory, including reasoning information highlighting past experiments. Consequently, the input prompts vary across rounds, with later rounds incorporating more accumulated information.
>
> To further address your comment, we describe how the output rules evolve throughout the optimization rounds on the electricity dataset. Due to space constraints, we show the first and last five output rules below. In the early optimization stages, the LLM produces diverse outputs, suggesting active exploration of possible rules. In contrast, during the later stages, the LLM refines the solution space around previously discovered solutions, making only minor adjustments. We hope this additional analysis provides deeper insight into our method.
>
> First five:
> - x12 = x3 + (x5 * x8) * x2 - x6
> - x12 = x5 * x2 + (x3 * x4 - x6) - x1
> - x12 = x11 + (x7 * x10) * x2 - x8
> - x12 = x3 + (x6 * x8 * x2) - x1
> - x12 = (x1 * x6 + x2 * x8) * x3 - x7
>
> Last five:
> - x12 = ((x2 - x1) * (x11 - x4)) ** 2 - (0.5 * (x1 - 0.3))
> - x12 = ((x2 - x1) * (x11 - x4)) ** 6
> - x12 = ((x2 - x1) * (x11 - x4)) ** 14 - (0.02 * (x5 - x7)) - 0.5
> - x12 = (x11 - x1) + (x11 * (x6 - x7))
> - x12 = ((x2 - x1) * (x11 - x4)) ** 3 + (0.1 * (x5 - x6)) ** 2
>
> ---
> **[W2-2, Q4] Handling hallucinations and their frequency across different LLMs.**
>
> While it is true that the LLM can occasionally suggest suboptimal or semantically incoherent rules, our method is designed to overcome these hallucinations. Specifically, we provide feedback on previously generated rules to guide the LLM to iteratively improve its rule generation. This feedback loop helps the LLM avoid hallucinations, which have likely resulted in low validation scores in subsequent iterations. Empirically, we have found that these issues occur more frequently in the early stages when the LLM explores the rule space more extensively and for less capable LLMs, e.g., those without additional training on dialogue or code generation data.
>
> ---
> **[Q3] Case studies for different LLMs (e.g., results, patterns).**
>
> In our ablation study, we evaluated three different LLMs. As shown in Table 3 of our manuscript, our custom model (the Llama 2 trained additionally on a high-quality dialogue dataset) achieved the lowest average error rate, followed by the Code Llama and then the Llama 2 chat base, which still outperformed the baseline XGBoost. In terms of feature generation patterns, we observed several differences between the models.
>
> Code Llama: This model utilizes various numpy operations (e.g., `np.sin`), which can generate mathematically sophisticated features.
> - x9 = ((x4+0.24) * x1) + ((x5+0.27) * x2)
> - x9 = np.sin(x1) * np.cos(x4)
> - x9 = x4 * np.tan(x8)
>
> Llama 2 chat base: This model favors various polynomial combinations.
> - x9 = x4 * x1 * (x2 + x6) ** 2
> - x9 = x4 * x1 ** 3 * (x2 + x6) ** 4 * (x7 + x8)
> - x9 = x4 * x1 ** 2 * (x2 + x6) ** 3 * (x7 + x8) * (1 + x5 ** 4)
>
> Ours: Our custom model also tends to explore polynomial space of features, often using built-in Python functions such as `abs()`. However, our model explores a considerably broader space of features more effectively than the Llama 2 chat base.
> - x9 = x1 ** 2 * (x2 - x3)
> - x9 = abs(x1) ** x1 + x2 - x3 - x4 - x5 - x6 - x7 - x8
> - x9 = x1 * (x1 + 2) - x2 * (x2 - 0.5) * (x2 - 0.5)
>
> These observations suggest that while all three models have strengths in feature generation, our custom model's broader exploration and more effective use of the feature space contribute to its superior performance. However, we believe equipping our custom model with enhanced code generation capabilities, such as utilizing scientific computing libraries like numpy, could lead to even more performance gains. We will incorporate these points into our final manuscript.
>
> ---
> **[Q6-2] Figure 3: Details on the dataset and evaluation with various metrics.**
>
> The dataset used in Figure 3 is nearly balanced with a class distribution of 55:45. Since it is a binary classification, the model outputs the class with the higher probability as the prediction result (i.e., no cut threshold). While accuracy is a suitable metric, we also report the ROC-AUC, a threshold-independent metric, in Figure 1 in the attached PDF to provide additional insights. Here, we highlight that using the real values improves all metrics, demonstrating that OCTree effectively recommends useful columns for the target task.
>
> ---
> **[Q6-1, 7] Editorial comments on Figure 3 and Table 7.**
>
> Thanks for your feedback. In the final draft, we will revise Figure 3 to be monochromatic (as in Figure 1 in the attached PDF) and make the best results bold in Table 7.

---

> > ### Comment · Reviewer_6JM8 · 2024-08-12
> >
> > We appreciate the authors' effort in answering our questions and the other reviewers' questions. We have no further questions and have decided to update our scores and slightly increase the final score.

---

> > > ### Author Response · Authors · 2024-08-13
> > > **Thank you very much for the response**
> > >
> > > Dear reviewer 6JM8,
> > >
> > > Thank you very much for letting us know! We are happy to hear that our rebuttal addressed your questions well.\
> > > If you have any further questions or suggestions, please do not hesitate to let us know.
> > >
> > > Thank you very much,\
> > > Authors

---

### Author Rebuttal · Authors · 2024-08-07

Dear Reviewers and ACs,

We sincerely appreciate the time and effort you dedicated to reviewing our manuscript and providing insightful comments. Below, we discuss some of the common points made by reviewers.

---
**Comparison with CAAFE [1].**

**Restricted applicability of CAAFE.** We would first like to clarify that, as mentioned in Section 4.1 of our manuscript, we originally excluded CAAFE from our comparison as it requires a language-based context for the dataset. This requirement limits CAAFE's applicability in cases where language descriptions are not explicitly provided, such as when feature names and values are obfuscated to protect confidentiality, as is common in financial and medical datasets. In contrast, the methods that we evaluated in Table 4, as well as our approach, are context-agnostic and can be applied to datasets without such linguistic descriptions. However, our method does benefit from clear language descriptions, as shown in Table 1, when available.

**Comparison with CAAFE.** To further address the reviewers’ concerns, we conducted a comparative evaluation using all datasets with contextual information from the experiments summarized in Table 1 of our manuscript. First, we would like to note that CAAFE showed high variance, even on the same dataset split, maybe due to the randomness of GPT-4o's temperature sampling (and sometimes failed to improve the baseline). Therefore, we first averaged the performance of three trials per random split. Then, we report the mean and variance of the three random splits. As shown in Table 1 of the attached PDF, our method consistently outperforms CAAFE (note that we slightly modified the official code of CAAFE to implement regression tasks). Notably, our approach using the open-source Llama 2 model fine-tuned on dialogue data outperforms CAAFE even when it employs the GPT family of models.

**Why is our approach better?** First, we would like to clarify the key distinctions between CAAFE and OCTree. Our approach generates much more semantically meaningful column names (e.g., ‘Smoking Status’), which serve as the basis for creating additional, high-quality columns. By leveraging the LLM’s reasoning, optimization, and in-context learning capabilities, we provide a history of validation scores for candidate features, along with decision tree reasoning, to guide the LLM in effectively navigating the feature space and generating relevant and coherent rules. In contrast, CAAFE primarily relies on the LLM’s language understanding capabilities to suggest simple combinations of existing features, such as binary feature crosses. CAAFE also tends to add new features in a greedy manner, evaluating the validation score for a candidate feature once and discarding it immediately if there is no improvement. Our approach, however, iteratively optimizes each candidate feature, leading to a more effective exploration of potential features.

**Case study.** As illustrated in the example below, CAAFE fails to introduce meaningful columns for the disease dataset, while our method proposes more relevant and coherent rules.

CAAFE:
- df['Age Category'] = pd.cut(df['Age'], bins=[0, 30, 60, 100], labels=['Young', 'Adult', 'Senior'])
- df['Fever_Cough_Interaction'] = df['Fever'] * df['Cough']

Ours:
- If the individual has “Fever” and “Fatigue” and “Difficulty Breathing” with “Age” between 60 and 90, then predict “Exposure to Infected Individuals” as “Yes”. Otherwise, predict “Exposure to Infected Individuals” as “No”.

**Additional advantages of OCTree.** Moreover, the semantically meaningful features generated by our method enable human annotators to provide additional labels, which can significantly enhance model performance. For instance, as shown in Figure 3 of our manuscript, collecting actual values for the suggested feature substantially improves patient mortality prediction.

**Combining OCTree with other automatic feature engineering methods.** Lastly, we want to highlight that our method is complementary to some of the existing automatic feature engineering techniques. As demonstrated in Table 4 of our manuscript, combining our method with OpenFE [3] leads to further performance improvements. Thus, it is also possible to first generate features with our method and subsequently employ CAAFE to further enhance the feature set.

We thank all the reviewers for the valuable questions and suggestions and will incorporate the results in the final manuscript.

---
**All References.**

[1] Hollman et al., LLMs for Semi-Automated Data Science: Introducing CAAFE for Context-Aware Automated Feature Engineering, NeurIPS 2023.

[2] Horn et al., The autofeat Python Library for Automated Feature Engineering Selection, ECMLPKDDW-ADS 2019.

[3] Zhang et al., OpenFE: Automated Feature Generation with Expert-level Performance, ICML 2023.

[4] Wang et al., Group-wise Reinforcement Feature Generation for Optimal and Explainable Representation Space Reconstruction, KDD 2022.

[5] Wang et al., Reinforcement-Enhanced Autoregressive Feature Transformation: Gradient-steered Search in Continuous Space for Postfix Expressions, NeurIPS 2023.

---

### Decision · Program_Chairs · 2024-09-25

**Decision:**

Accept (poster)

**Comment:**

This authors propose a new tabular learning framework called OCTree (Optimizing Column Feature Generator with Decision Tree Reasoning) that leverages the reasoning capabilities of large language models (LLMs) to automatically generate new column features based on feedback from decision trees (i.e., it is an automatic feature space construction method). Methodologically, OCTree algorithm uses LLMs to recommend new features to be generated by name and feature generation rule, which is then used in the training of the ML algorithm for validation. Experimental results demonstrate that OCTree consistently outperforms existing automatic feature engineering methods across various tabular data prediction tasks.

Consensus strengths of the paper identified by reviewers (and myself) include:
- The idea of using LLMs to generate novel features for ML models over tabular data is conceptually appealing and the specific approach proposed does this in a straightforward way (meant as a positive).
- The paper is very well-written, well-structured, and the empirical results are promising.
- The experiments have a sufficient number of LLMs, baseline algorithms, strong ML models for these tasks, benchmark datasets, and ablation studies to support the core statements claimed in the paper. I actually don't think the reviewers sufficiently appreciate how difficult XGBoost is to beat "out of the box" for these tasks -- which it is so widely used.

Additionally, reviewer concerns regarding comparison to CAAFE and scalability issues were sufficiently addressed during rebuttal.

Conversely, consensus (and my own) observed weaknesses of this work include:
- While issues regarding CAAFE and scalability were addressed during rebuttal, they still need to be integrated into the paper -- but this seems possibly just by editing a revised manuscript.
- Some experimental details (e.g., number of rounds, inter-round feedback) aren't easy to find in the paper (but are in the rebuttal). However, this is easily added.
- While the number of datasets is sufficient, the paper would be non-negligibly strengthened by more datasets, etc. as this would potentially show utility for different cases and could provides cases where interpretability, etc. are more salient (and in line with motivate). Honestly, this would make me more confident in recommending a clear accept.
- While I don't think that the existing experiments were 'polluted' by memorized features (i.e., reviewer 245r), this isn't too difficult to test for (at least for the Kaggle workshop cases) but could be a problem in competition settings -- but likely not an issue in actual deployed systems. Thus, I would include some discussion of this.
- In the same vein, i would add that the generated features are more interpretable than other feature space augmentation algorithms. Of course, generated features are often complex and thus formal interpretation is difficult, but the LLM labels and rules may be useful for this. Discussion is worthwhile as tabular data domains often have this aspect.
- I do agree with reivewer Pmbp that discussion regarding SDP-based feature generation methods is worthwhile; I don't believe empirical comparisons are needed, but this also occurred to me while reading.
- The discussion regarding hallucination in rebuttal should be addressed in the paper. It is clear to me that the ML algorithms can likely perform feature selection (as they have been doing for a long time), but should probably be mentioned.

Overall, I believe this is a nice paper that may have solid practical implications in many 'traditional ML' domains. Additionally, as LLMs continue to improve (including reasoning), I expect this to inspire future work. While there are some details that should be added to the paper, it is well-written overall and shows good performance and convincingly supports the core claims of the work.